# Development of Serotonergic and Dopaminergic Neuronal Networks of the Central Nervous System in King Crab, *Paralithodes camtschaticus*

**DOI:** 10.3390/biology13010035

**Published:** 2024-01-08

**Authors:** Elena Kotsyuba, Arman Pahlevaniane, Sergei Maslennikov, Vyacheslav Dyachuk

**Affiliations:** A.V. Zhirmunsky National Scientific Center of Marine Biology, Far Eastern Branch, Russian Academy of Sciences, Vladivostok 690041, Russia; epkotsuba@mail.ru (E.K.); armanpahlevanyan@gmail.com (A.P.); 721606@mail.ru (S.M.)

**Keywords:** serotonin, dopamine, synapsin, larval development, metamorphosis, red king crab

## Abstract

**Simple Summary:**

This study provides the first description of the pattern of central nervous system development in red king crabs, in particular the changes that take place during the two metamorphoses of their free-swimming planktonic larvae. As a result of the study, a map of the neural architecture and distribution of cells producing serotonin and dopamine in the medial brain and ventral nerve cord (VNC) of the larval nervous system has been obtained, which gives insight into the potential functions of serotonin and dopamine both in red king crab larvae at all developmental stages and in adults.

**Abstract:**

This article presents recent findings as regards distribution of cells producing serotonin and dopamine in the larval central nervous system at different developmental stages, including four pelagic larval stages (zoea I–IV), a semibenthic postlarval stage glaucothoe (megalopa), benthic juveniles, and adult red king crabs, *Paralithodes camtschaticus*, made by using immunocytochemistry and confocal laser scanning microscopy. We have shown that the serotonergic and dopaminergic neurons are present long before the onset of metamorphosis. In the red king crab b larval nervous system, the changes become particularly pronounced during the first metamorphosis from zoea IV to glaucothoe, which may be related to the development of the segmental appendages and maturation of motor behaviors in decapods. This work presents the distribution and dynamics of the development of serotonergic and dopaminergic neuronal networks in king crab show, the potential roles of serotonin and dopamine in the modulation of olfactory and visual processing in the early stages of larval development, and also the mechanosensory and chemosensory processing in the glaucothoe stage during settlement and in their transition from a pelagic to benthic lifestyle.

## 1. Introduction

Free-swimming planktonic larvae are a key stage in the development of many marine phyla, and, therefore, studies of these organisms contribute much to our understanding of major genetic and evolutionary processes. However, our knowledge of the ontogenetic processes that shape the crustacean nervous system is very limited to date. Larval development at the immature stages of bottom-dwelling decapod crustaceans has been investigated in sufficient detail only for major brachyurans species, while anomurans still remain poorly studied, despite their high diversity.

King crabs (Lithodidae Samouelle, 1819) comprise a group of large anomuran decapods found mainly in the cold waters of the Pacific and Atlantic Oceans [1]. The red king crab *Paralithodes camtschaticus* (Tilesius, 1815) is one of the most commercially valuable crustaceans and is also an important species in aquaculture [2]. It has a complex life cycle with multiple larval stages, including four pelagic larval stages (zoea I–IV) and a semibenthic postlarval stage glaucothoe (megalopa), followed by benthic juvenile and adult stages [3]. In decapod crustaceans, zoea larvae possess a wide range of organ systems necessary for surviving and developing in the pelagic environment, including a digestive system, osmoregulatory organs, neuromuscular system, and sensory organs to detect environmental cues [4]. They live in the water column and make vertical diurnal migrations, and can respond to changes to environmental factors: light, hydrostatic pressure, tidal currents, temperature, salinity, and food concentration [5]. Additionally, when decapod crustacean larvae are confronted with unfavorable salinity conditions, they may also respond with an active avoidance behavior. The larval development in king crabs is characterized by a double metamorphosis: the first metamorphosis between the zoea IV stage and the glaucothoe; the second between the glaucothoe and the first juvenile stage, when the transition from a pelagic to a benthic life style occurs [5,6]. These processes require numerous modifications in behavioral patterns and morphology [7], and are accompanied by changes in their nervous system [8].

To date, the structure and development of the CNS within the Decapoda have been described from several brachyuran decapod species [8,9,10,11,12,13,14] by various methods, including confocal laser scanning microscopy (CLSM) and microcomputed X-ray tomography [4,7]. Anomurans, although having a comparable life history that involves a series of zoeal larvae followed by a single megalopa stage, differ from brachyurans in the details of their developmental stages and exhibit a distinct, species-specific nervous system architecture [15]. Although advances have been made to elucidate the anatomy of anomurans’ nervous system at certain developmental stages [12,13,15], there remains a poor understanding of the neurotransmitter specialization of special types of neurons, as well as the development of neurotransmitter systems and their role in larval development stages and adult crabs. As has been shown for many species of aquatic invertebrates, serotonergic cell neurons are the first to differentiate and represent the central component of the larval nervous system [16,17,18].

In decapod crustaceans, as in vertebrates and other invertebrates, the neurotransmitters dopamine (DA) and serotonin (5-hydroxytryptamine, 5-HT) are well known, not only as modulators of a wide range of neurophysiological processes, but also as the earliest modulators to appear and be widely distributed in developing neural circuits [16,17,18,19,20,21,22,23]. Several researchers have demonstrated the presence of 5-HT and DA in the embryonic and larval nervous systems of decapods [8,18,19,20,21]. These neurotransmitters are known to influence every single developmental process, but are particularly essential in larval development, when activity-dependent mechanisms sculpt neural circuits in response to environmental changes and are maintained until adulthood. However, to date, little is known about the localization and expression patterns of 5-HT and DA in the nervous system during larval development and metamorphosis, because usually only one particular developmental stage has been considered, whereas changes during development in crabs are neglected.

Therefore, in the present study, we performed immunohistochemical labeling and examined it using CLSM to elucidate the distribution of 5-HT and DA-immunoreactive neurons in the CNS of *P. camtschaticus* larvae (from zoea I to the first juvenile stages) and adults. We provide here the first description of the serotonergic and dopaminergic neuronal networks in *P. camtschaticus*, which will provide a more comprehensive knowledge of the king crab larval nervous system and improve our understanding of the neural mechanism of behaviors at different developmental stages.

## 2. Materials and Methods

### 2.1. Animals

Larvae of the red king crab *P. camtschaticus* for this study were reared in an experimental system of pools for culturing red king crab, installed at the Zapad Marine Biological Station, A.V. Zhirmunsky National Scientific Center of Marine Biology, Far Eastern Branch, Russian Academy of Sciences (Vostok Bay, Sea of Japan). The system was designed by experts in the artificial rearing of red king crab. The rearing protocol was described in detail by Anger and Nair [24]. The larvae were kept in 400 L basin complexes; their planting density was 50 individuals per 1 L. The larvae were kept at a constant temperature of 7–8 °C, a salinity of 30–31‰, a dissolved oxygen concentration of 8.1–8.5 mg/L, and under a natural light/dark cycle of 12: 12 h, and fed brine shrimp nauplii, *Artemia* sp., 2 times a day.

The larval development under these conditions is stable, and the larval stage, age, and even molt cycle can be accurately identified. After the emergence of zoea I, the pools were daily inspected for molts in order to determine the duration of each stage. Each larval stage (zoea I, zoea II, zoea III, zoea IV, glaucothoe, and juveniles) was sampled at the intermolt (i.e., when 50% of each molt cycle had occurred). The characteristics and duration of the larval stages at a specified temperature (7–8 °C) are presented in Table 1. All possible measures were taken to minimize the number of animals used in this study.

Larvae of each stage were selected, relaxed for 5–10 min in a MgCl_2_ solution, and fixed in a 4% paraformaldehyde solution in phosphate-buffered saline (PBS; 100 mM Na_3_PO_4_, 140 mM NaCl, and a pH of 7.4) for 2 h at room temperature. Ten specimens of each stage were used as whole mounts; another ten specimens of each stage were processed for the cutting sections. The fixed larvae were washed in 0.1 M PBS. For the whole mounts, the specimens were dehydrated in a series of graded methanol concentrations (25, 50, 75, and 100%) and stored in 100% methanol at −20 °C.

Adult male red king crabs measuring 150 mm in carapace width were captured in Peter the Great Bay, Sea of Japan. The animals were then kept in tanks with aerated seawater at a temperature of 7–8 °C, a salinity of 30–31‰, and a dissolved oxygen concentration of 8.1–8.5 mg/L under a natural light/dark cycle of 12: 12 h. During the 2-week period of adaptation, the water in the tanks was changed three times a week, and the animals were fed fresh blue mussels (*Mytilus edulis*) once a day.

Prior to fixation, the crabs were relaxed for 5–10 min in a MgCl_2_ solution. Then, their nervous ganglia were immediately dissected and fixed in a 4% paraformaldehyde (PFA) solution in PBS (100 mM Na_3_PO_4_, 140 mM NaCl, and a pH of 7.4) for 2 h at room temperature. Then, the samples were washed with PBS (3 × 20 min) and incubated overnight in 30% sucrose (prepared in PBS) at 4 °C for cryoprotection. Afterwards, the samples were embedded in a freezing medium (OCT cryomount, HistoLab, Sweden) and frozen for storage at −20 °C. Tissue cryosections of 30 μm were then cut on an HM525 cryostat (Thermo Fisher Scientific, Waltham, MA, USA).

These sections were mounted on Superfrost Plus microscope slides (Thermo Fisher Scientific, Waltham, MA, USA), air-dried, and stored at −20 °C for subsequent staining.

In all the zoeal stages, the carapace length was measured from the rostrum base to the posterior edge of the carapace without posterior spines. In the glaucothoe and juvenile stages, the carapace length was measured from the eye notch to the posterior edge of the carapace. The measurements were expressed as the arithmetic mean (and standard deviation) of the carapace measurements taken during this study.

### 2.2. Immunohistochemistry

For the immunohistochemical staining of the ganglia from the adult animals, the freshly frozen sections were processed as described previously [25]. Immunostaining was used to determine the distribution of serotonin or tyrosine hydroxylase (TH), the enzyme that catalyzes tyrosine into DOPA, the precursor of DA. To visualize the outline of the neuropil structure, immunostaining with antisynapsin antibodies was also performed. To eliminate nonspecific binding, the slides were incubated overnight in a blocking buffer consisting of 10% normal donkey serum (NDS) (Jackson ImmunoResearch, Cambridgeshire, UK), 1% Triton-X 100, and 1% bovine serum albumin (BSA; Millipore, Burlington, MA, USA) dissolved in 1× PBS at 4 °C. Additionally, we dissolved the following polyclonal primary antibodies in this buffer: rabbit anti-TH (1:1000; Millipore), rabbit or goat anti-5-HT (1:1000; ImmunoStar Inc., Hudson, WI, USA), and mouse antisynapsin antibody (1:1000; 3C11 (anti SYNORF1); Developmental Studies Hybridoma Bank, Iowa City, IA, USA). Subsequently, the sections were washed in 0.01 M PBS (pH of 7.4) containing 0.5% Triton X-100 (pH of 7.4) prior to incubation with 488-, 555-, or 647-Alexa Fluor-conjugated donkey secondary antibodies (1:1000; Invitrogen, Waltham, MA, USA) along with the nuclear marker 40,6-diamidino-2-phenylindole (DAPI; Sigma-Aldrich, St. Louis, MO, USA) for 2 h at 22 °C. The sections were then washed with PBS and embedded in glycerol (Merck, Kenilworth, NJ, USA).

For the immunohistochemical staining of the larvae, we used the previously described whole-mount immunostaining protocol [26]. Larvae of each stage, stored in 100% methanol, were transferred to 0.1 M PBS by changing the methanol solutions with decreasing concentrations. The samples were incubated overnight in a 1% ethylenediaminetetraacetic acid (EDTA) solution in PBS at room temperature for decalcification. The specimens were rinsed in PBS supplemented with 0.1% Triton X-100 (PBST) for 4.5 h with agitation. Then, the specimens were incubated overnight in a blocking solution (10% donkey normal serum, 1% bovine serum albumin, and 1% Triton X-100, 0.003% NaN_3_) in 0.1 M PBS.

For the detection of the nerve structures, the larvae were incubated with primary antibodies: serotonin goat or rabbit polyclonal antibody (ImmunoStar, 20079 and 20080, ImmunoStar Inc., Hudson, WI, USA), tyrosine hydroxylase (TH) rabbit polyclonal antibody (AB152, Millipore), and monoclonal mouse anti-synapsin antibody (AB528479, DSHB) in the blocking solution at a dilution of 1:1000 for 5 days at 4 °C. Then, after being washed in PBS (4 × 20 min), the samples were incubated overnight at 4 °C in donkey anti-goat (A32814, Invitrogen, Thermo Fisher Scientific), donkey anti-rabbit (A32794, Invitrogen, Thermo Fisher Scientific), and donkey anti-mouse (A32787, Invitrogen, Thermo Fisher Scientific) antibodies at a dilution of 1:1000 with 0.1 μg/mL DAPI. The larvae were then washed in PBS with 0.1% Tween 20 (PBST) (5 × 20 min). All specimens were prepared for confocal microscopy and mounted on glass slides in a drop of 70% glycerol. For the controls, we showed that the preincubation of the 5-HT antibody with the same conjugate (10 μg/mL, 20080, ImmunoStar Inc.) at 4 °C overnight eliminated all of the immunolabeling of serotonin in the tissues. The preadsorption of the diluted rabbit/goat antiserum with 10 mg/mL bovine serine albumin (BSA) overnight at 4 °C did not influence this staining, i.e., these antibodies recognized only serotonin and not BSA. Also, only secondary antibodies (without the treatment of primary antibodies) were used as the control antibodies for the crab larvae.

### 2.3. Specification of the Primary Antibodies

We used polyclonal rabbit or goat antibodies that targeted BSA-bound 5-HT with paraformaldehyde (20080 and 20079, respectively; ImmunoStar Inc.). According to the manufacturer’s instructions, staining with these antibodies was completely eliminated upon pretreatment with 25 mg of the 5-HT–BSA conjugate per 1 mL of diluted antibody. We demonstrated that the overnight preincubation of the antibody with 10 mg/mL of the conjugate (20080, ImmunoStar Inc.) at 4 °C completely eliminated the 5-HT immunolabeling in the control tissues. Furthermore, the overnight preadsorption of the diluted antibody with 10 mg/mL BSA at 4 °C did not affect this staining (i.e., these antibodies recognized 5-HT alone and not BSA). This 5-HT antibody was previously used to detect 5-HT in the brains of arthropods, including crabs, hermit crabs, and lobsters [27,28]. The rabbit anti-tyrosine hydroxylase (TH) antibody (AB152; Millipore) targets TH as a key enzyme involved in tyrosine biosynthesis. The anti-TH antibody was previously identified in the ganglia of arthropods, including crustaceans [29], thereby validating the use of a TH antibody as a reliable marker of dopaminergic neurons. To visualize the outline of the neuropil structure, these sections were incubated with mouse monoclonal anti-synapsin antibody which targets a presynaptic marker (SYNORF1 or antibody 3C11). A previous study showed that this antibody detects an epitope widely conserved in the nervous system of arthropods, including crustaceans [30,31]. The sections were then preincubated with the secondary antibody and incubated with 10 mg/mL of the nuclear marker DAPI (in PBS).

### 2.4. Confocal Microscopy and Imaging

The samples of the immunocytochemically stained larvae and the histological sections were scanned on an LSM 780 confocal microscope (Zeiss, Jena, Germany) using the Zen software version 3.6 with the following laser wavelengths: 405, 488, 555, and 647 nm. All the larvae images were composed in the Z-stack mode with an optical slice thickness of 1 μm along the *Z*-axis. The obtained images were transformed into projections in the maximum intensity mode. The resulting images were processed using the IMARIS 7.0 software (Bitplane, Switzerland). The converted images were saved as TIFF images and transferred to the Photoshop CS software (Adobe, San Jose, CA, USA), where their contrast and brightness were adjusted for optimal clarity. Negative controls for each fluorochrome were scanned with the same settings. The final figure panels were edited in Adobe Illustrator CS2 (Adobe System, San Jose, CA, USA).

### 2.5. Terminology

The neuroanatomical nomenclature used in this study is based on that proposed by [12], with some modifications according to [10,32]. Also, when describing the general morphology of the larvae, we used the anatomical classification as proposed in publications by [4]. To describe the anatomy, neuronal cell clusters, and the ganglionic neuropils of the adult *P. camtschaticus* brain, we used the standard nomenclature developed in previous studies on decapods [10,14,30,31,33]. In the present study, we did not consider the neuropils located in the eyestalk, including a part of the protocerebrum (hemiellipsoid body and terminal medulla).

## 3. Results

### 3.1. Synapsin-like Immunoreactivity

The CNS in adult and larval *P. camtschaticus* is comprised of a brain and a ventral nerve cord (VNC), including fused subesophageal ganglia (mandible, maxilla 1 and maxilla 2, and three maxilliped ganglia), five thoracic ganglia associated with the pereiopods (walking legs), and six pleonic (abdominal) ganglia (Figure 1A–G). The brains and VNCs are connected via a pair of circumesophageal connectives extending from the tritocerebrum running through the commissural ganglion (CG) (Figure 1A–C,F,G). As in other decapods, the brain in *P. camtschaticus* is divided into three parts: a protocerebrum, a deutocerebrum, and a tritocerebrum, with each containing its own neuropils and associated neuronal cell clusters. The protocerebrum is subdivided into the optic ganglia and the lateral and median protocerebrum. Both the optic ganglia and the lateral protocerebrum are located in the eyestalk, while the median protocerebrum, deutocerebrum, and tritocerebrum make up the median brain (Figure 1A–C).

During the development from zoea I to the first juvenile stage, the size of the median brain was observed to gradually increase. In zoea I, the breadth of the brain measured in a transverse section at the level of the center of the olfactory neuropils was ~280 μm, then it increased to ~300 μm in zoea II, to ~345 μm in zoea III, to ~420 μm in zoea IV, to ~560 μm in glaucothoe, and reached ~600 μm at the first juvenile stage.

In the median brain, synapsin-like immunoreactivity (Syn-LIR) was detected in the median protocerebrum, deutocerebrum, and tritocerebrum (Figure 2A–G). During the larval development, marked changes were recorded from the olfactory neuropils (ONs) of the deutocerebrum (Figure 2A–F). The formation of nonhomogeneous microglomeruli occurred as early as in zoea I, but the ON had the cylindrically shaped glomeruli around their periphery only at the later developmental stages and in adult *P. camtschaticus* (Figure 2G). In glaucothoe, the olfactory neuropil showed a tendency to be subdivided into three segments (Figure 2F). Furthermore, small accessory neuropils (AcNs) were present in adult *P. camtschaticus*, being located close to the origin of the olfactory globular tract (Figure 2H). In the tritocerebrum at all stages, the large antenna 2 neuropil (AnN) displayed strong Syn-LIR, but no transversely stratified or segmented pattern was observed within the antenna 2 neuropil, as described from some species of anomurans and brachyurans [10,31].

In the VNCs at all larval stages of *P. camtschaticus*, the fused ganglia of the first six segments (the mandible, maxilla 1 and maxilla 2, and three maxilliped ganglia) formed the subesophageal ganglia (SEG) (Figure 1A–G and Figure 3A–E). During the development from zoea I to the first juvenile stage, the SEG exhibited a similar morphology and gradually increased in size. The breadth of the SEG measured in a cross-section near the base of the maxillules was ~180 μm in zoea I and increased to ~220 μm at the first juvenile stage. The thoracic ganglia (TG) (ganglia of pereion segments 1–5) showed a lower degree of fusion than the neuromeres of the SEG (Figure 1D,F,G and Figure 3A–G). In the development period from zoea I to the first juvenile stage, these were observed to gradually increase. During the first metamorphosis from zoea IV to glaucothoe, a pronounced lateral extension of the neuropil was present that corresponded to the first TG (Figure 3D–F). In zoea I, the neuropil breadth measured in a sagittal section through the lateral part of the VNC at the level of the first TG (the left half of the ganglion) was ~100 μm, increased to ~110 μm in zoea II, to ~130 μm in zoea III, to ~180 μm in zoea IV, to ~220 μm in glaucothoe, and reached ~230 μm at the first juvenile stage.

The abdominal ganglion (AG) (ganglia of pleon segments 1–6) was smaller than the thoracic ones and was identified as separate neural centers in the larval stages (Figure 1D,E). In the zoea, the abdominal ganglia 2 to 6 were not fused longitudinally and exhibited long intersegmental connectives. The first of the six abdominal ganglia was attached to the fifth thoracic ganglion. From the glaucothoe onwards, this ganglion was distinguished well between the fifth TG and the second AG (Figure 3E,F,H). After the first metamorphosis, the AG in the glaucothoe became shorter than in zoea IV (Figure 1A,B). In the first juvenile stage and in adult crabs, they were relatively small and fused with each other and with the fifth TG (Figure 1A and Figure 3H).

### 3.2. Serotonin-like Immunoreactivity

In larval and adult *P. camtschaticus*, 5-HT-LIR immunoreactive neurons and nervous fibers could be identified. Their distribution and pattern of emergence in different CNS regions is presented in Figure 4, Figure 5, Figure 6, Figure 7 and Figure 8.

In the median protocerebrum, 5-HT-LIR was detected in cell bodies lying dorsoanteriorly to the medial protocerebral neuropils in both larvae and adult crabs (Figure 4A–C). This cell cluster was located between the protocerebral tracts (PT) in the same position as the anterior dorsal cell cluster (ADC) 6 in adult crabs (Figure 4F), which suggests that they are homologous in the larval and adult brains. In zoea III, 5-HT-LIR neurons of 25–30 μm were found ventrally of the ADC (Figure 4C). In the glaucothoe, similar immunolabeled neurons were observed between the anterior medial protocerebral neuropil and the olfactory neuropils at the level of the central body in cluster 8 (Figure 4D,E). In the medial protocerebrum, single 5-HT-LIR fibers were identified already in zoea I, but the most intense immunoreactivity was detected in the anterior and posterior medial protocerebral neuropils (AMPN and PMPN), the protocerebral bridge neuropil (PB), and the central body (CB) from the glaucothoe stage onwards (Figure 4A,D,E).

In the deutocerebrum of zoea II, large 5-HT-LIR neurons were detected medially of the olfactory neuropils (Figure 5A). From glaucothoe onwards, this group included also medium-sized (20–30 μm) and small (10–15 μm) 5-HT-LIR neurons (Figure 5B–F). The neurons of this group were located in the same position as cluster 11 (dorsolateral) in adult crabs. In some of the mounts, there were branches of large 5-HT-LIR neurons, the so-called dorsal giant neurons (DGN), according to the classification by Sandeman et al. [34], which innervate the glomerulus of the olfactory neuropils (Figure 5D). In glaucothoe, 5-HT-LIR neurons were found in cluster 11 that sent neuronal fibers to the median antenna I neuropil (MAN) (Figure 4E). Furthermore, glaucothoe had small (12–18 μm) 5-HT-LIR neurons located in the same position as cluster 9 (anteroventral) (Figure 4D and Figure 5E). In adult crabs, 5-HT-LIR was also detected in small (10–12 μm) globular neurons of cluster 10, located posteriorly of the olfactory neuropils in the dorsal region (Figure 5F). In the olfactory neuropils, 5-HT-LIR fibers were detected from zoea II to the first juvenile stage and in adult crabs. Small-sized 5-HT-LIR neurites were present throughout the area of glomeruli that were arranged on the periphery of the olfactory neuropils (Figure 4C and Figure 5A–D,F). Large 5-HT-LIR fibers were detected predominantly in the central part of the olfactory neuropils (Figure 5F). In the small accessory neuropils of adult crabs, 5-HT-LIR was not found. In the tritocerebrum, 5-HT-LIR fibers were present in the antenna II neuropils (AnN) (Figure 5C). The antenna 2 neuropil was big, extended towards the antenna 2 nerve, and showed a clear 5-HT-LIR. Furthermore, 5-HT-LIR fibers were found also in the esophageal connectives (OCs) (Figure 6A,B). The latter connected the tritocerebrum with the commissural ganglia (CG), forming a network there (Figure 6B), and then extended farther along the VNC.

In the VNC, most 5-HT-LIR was located in the neurons of the SEG (Figure 6C–F and Figure 7A–F,H). At all the larval stages from zoea I to the first juvenile (crab 1) stage, SEG 1–6 showed a repetitive pattern of dorsolaterally situated 5-HT-LIR neurons that are referred to as anterior serotonergic cells in accordance with the terminology by Harzsch, Waloszek [27] (Figure 6C–F and Figure 7A–F). These 5-HT-positive cells were present in each neuromere of the SEG. Furthermore, in the SEG, single 5-HT-LIR cells, referred to as posterior serotonergic cells according to Harzsch, Waloszek [27], were localized near the midline (Figure 7C,E,H). Processes of these neurons and 5-HT-LIR fibers that ran along the midline in the VNC formed a heavily stained plexus of varicosities in the neuropil at a level extending from the first to sixth SEG (Figure 6D,E and Figure 7A–E). In the pereion (thoracic) ganglia in zoea II, single anterior serotonergic neurons with a low 5-HT-LIR intensity were detected in neuromeres 1–3 (Figure 6D). Furthermore, the pair of neurons in neuromere 4 showed a high 5-HT-LIR intensity in all the developmental stages (Figure 6D,E and Figure 7D–I). In zoea III and onwards, the staining intensity of the anterior serotonergic neurons in neuromeres 1–3 increased (Figure 7D,H,I). These segmental types of anterior neurons were present in each of pereion segments 1–5 at all the subsequent developmental stages. In zoea II, a small 5-HT-LIR neuron with the processes extending parallel to the pereiopod was found between neuromeres 2 and 3 (Figure 7F). Furthermore, zoea II had a medial 5-HT-LIR neuron in the thoracic neuromere 2, whose processes were found in the abdomen and near the digestive gland (Figure 8A). At all development stages, intensely stained pairs of posterior serotonergic neurons of 50 μm were present in neuromere 5 (Figure 6A,C–F, Figure 7A,F,I and Figure 8B,C,E,F). As was observed in some of the mounts, their primary neurites ran anteriorly until reaching the subesophageal neuropil (Figure 6D and Figure 8C).

In the first abdominal (pleonic) ganglion, 5-HT-LIR was detected in the pairs of neurons of 60 μm at all development stages (Figure 8B,D,E). At all the stages from zoea I to glaucothoe, a single pair of stained neurons was detected in the abdominal ganglia 2–6 (Figure 6C and Figure 8C,D,G,H).

### 3.3. Tyrosine Hydroxylase-like Immunoreactivity

At all larval stages and in adult *P. camtschaticus*, tyrosine hydroxylase-like immunoreactivity (TH-LIR) was present in cell bodies and fibers in the median brain and the VNC (Figure 9, Figure 10 and Figure 11). In the median brain of zoea I, a few TH-immunoreactive neurons were located in the anterior medial cell cluster 6 (Figure 9A). The number of labeled neurons in this cell cluster increased with development in zoea II–IV (Figure 9B). At all the larval stages and in adult crabs, TH-LIR was also observed in the fibers of the anterior and posterior medial protocerebral neuropils, and also in the fibers of the protocerebral tracts (Figure 9A,C,H).

In the deutocerebrum of zoea I, large TH-LIR neurons were found medially of the olfactory neuropils (Figure 9D). From zoea III onwards and in adult crabs, small (12–18 μm) TH-LIR neurons were also present in this group (Figure 9E,F). The olfactory neuropils of developing larvae showed low and moderate TH immunoreactivity. In adult *P. camtschaticus*, numerous small and large TH-LIR neurites of the olfactory neuropils were intensely labeled (Figure 9F,G). In adult crabs, TH-LIR was also identified in medium-sized neurons of cluster 13 (Figure 9G).

In the commissural ganglion (CG) of *P. camtschaticus*, TH-LIR cells were present at all the developmental stages. These were the so-called L-cells [29]. In *P. camtschaticus*, usually three cells were labeled in each commissural ganglion (Figure 9H,(H1) and Figure 10A,B). One TH-LIR cell appeared to bifurcate, sending one process towards the median brain and the other towards the VNC (Figure 9(H1)). Two more TH-LIR cells and their processes were located in close proximity to the TH-LIR fibers that extended to the ventral nerve cord (Figure 9(H1)). The TH-LIR fibers, termed as medial and lateral fiber bundles [20], ran in the longitudinal tracts all along the nerve cord from the CG to the last abdominal ganglion (Figure 10C,(C1) and Figure 11E). In the subesophageal ganglion in zoea I–IV, TH-LIR cells (ranging from 10 to 30 μm in diameter) were observed in the lateral aspects of the SEG (Figure 10D,E). In the glaucothoe, new TH-LIR cells were added in neuromeres of the subesophageal ganglion (Figure 10F). At the first juvenile (crab 1) stage, TH-LIR cells were identified in each subesophageal segment (Figure 11A,B). Furthermore, a medially located pair of large TH-LIR neurons was present in the first subesophageal ganglion (Figure 11C). In adult king crabs, a paired lateral group of TH-LIR cells, observed in the subesophageal ganglion, was dominated by cells of 60–70 μm (Figure 11D), which were not found in the larvae. In the thoracic (pereion) ganglia (TG) of zoea I–IV, TH-LIR cells were observed in thoracic segments 1 and 4 (T1 and T4) (Figure 10D and Figure 11F). At the first juvenile stage, TH-LIR cells (20 μm in diameter) were identified in each of the thoracic segments (Figure 11G). In the first juvenile stage and adult crabs, TH-positive cells (30 μm in diameter) and fibers were also observed medially (Figure 11A,B). In the first abdominal (pleonic) ganglia, TH-LIR cells (40–50 μm in diameter) were identified during the development from zoea I to juveniles and in adult king crabs. In some of the mounts, only one cell was stained, while two TH-positive cells were observed in the others (Figure 11C,E,H,I). The abdominal ganglia (AG 2–6) did not contain TH-immunolabeled cells, but thin TH-positive fibers were present in all the abdominal segments (Figure 9C,(C1) and Figure 10E).

## 4. Discussion

### 4.1. General Features of the Larval and Adult King Crab CNS

Our findings have shown that the general nervous system morphology in red king crab larvae is consistent with that previously described from representatives of other decapod species: *Porcellana platycheles* [15], *Carcinus maenas* [4,12,35], *Pachygrapsus marmoratus*, and *Hippolyte inermis* [15]. However, this study of the nervous system in a consistent temporal sequence in *P. camtschaticus* has revealed some noticeable changes in the architectures of the forming neuropils at specific developmental stages, in the median brain, and the VNC.

In the median brain, these major morphological changes are mostly associated with the development of olfactory neuropils, which are the primary sensory centers that process olfactory input [36,37].

In king crabs, the development of olfactory neuropils is characterized by not only an increase in their size, but also a tendency to form sublobes in these neuropils after the first metamorphosis. The presence of sublobes in the olfactory neuropil was also reported earlier for other anomuran species [30,31,33]. According to [33], the presence of sublobes in the olfactory neuropil may increase the surface area, thus providing a larger contact area between the enveloping plexus of the antennal nerves and the olfactory glomeruli. This feature is considered to be unique among anomuran decapods and is characteristic of hermit crabs and the Coenobitidae [30], which have a well-developed olfaction for better orientation in their habitat.

The other change in the larval nervous system in king crabs concerns processes of reorganization in the VNC during the first and second metamorphosis. In *P. camtschaticus*, there occurs a fusion of the SEG, TG, and (AG) ganglia (with condensed abdominal segments forming a tail). This contrasts to other palaemonid decapods, such as prawn (Caridea), lobster (Parinurids), and crayfish (Astacidea), where the SEG and TG are fused, but the AG remains separated from the other two structures [32,38], or in brachyuran crabs, where the AG is formed through the fusion of the significantly reduced abdominal segments, forming a very small tail [39].

Furthermore, differences between species have been found in the relative timing of the formation of certain neural elements. In red king crab, the processes of neuronal reorganization in the VNC become particularly pronounced at the first metamorphosis. At this time, maxillipeds 1, 2, and 3 lose their function, and their locomotor apparatus is represented by well-developed pleopods and is used for moving on the substrate. A comparison between various decapod taxa has shown that zoea I larvae in different species have different ganglia in the segments with well-developed limbs, being in a more advanced stage of differentiation compared to the segments with only limb buds or without externally visible limb anlagen [13,15]. In different crustacean species, the ganglia develop at different tempos, depending on general larval morphogenesis [40,41], which are related to the differences in larvae’s lifestyle, and to the development of their segmental appendages and the maturation of motor behaviors.

### 4.2. Serotonin-like Immunoreactivity

The 5-HT and DA are present at the early developmental stages in all decapod genera, suggesting that these ancient molecules play a vital role in the nervous system functions [8,19,20,27]. In the median protocerebrum at all larval stages and in adult king crabs, 5-HT-LIR is detected in the anterior medial cell cluster (cluster 6) of the median protocerebrum. Earlier, the presence of 5-HT-LIR in this group was reported for brachyurans, such as larvae of the spider crab *Hyas araneus* [8,13], and adult *Pacifastacus leniusculus*, *Procambarus clarkii*, *Cherax destructor*, *Scylla olivacea*, and *Litopenaeus vannamei* [42,43,44], and also for anomurans, such as adult *Coenobita clypeatus* and *Munida quadrispina* [30,45]. Moreover, 5HT-LIR somata in cluster 8 are already found in red king crab larvae at the early developmental stages. A similar class of serotonergic neurons (CBN4) with their somata and a respective projection pattern was described from crayfish [46]. This cluster 8 is reported to be innervated by serotonergic neurons [47] and implicated in a visual input pathway in crustaceans [48]. An increase in 5-HT-LIR during the larval development from zoea I to the first juvenile stage in the anterior/posterior medial protocerebral neuropils (AMPNs/PMPNs), central body (CB), and protocerebral bridge (PB) may be related to the increase in the number of 5-HT-LIR neurons in clusters 6 and 8 [46,49]. Protocerebral neuropils have been shown to be the location through which visual input information passes and finally integrates with the ON, in which chemosensory inputs from olfactory receptor neurons are processed [30,48]. In addition, some 5-HT- and TH-LIR neurons in cluster 6 might be involved in the visual pathway associated with the eye и, which might also indicate its involvement with the modulation of visual inputs in this red king crab.

The significant resemblance in the distribution of 5-HT-LIR in the neuropils of the median protocerebrum between larval and adult *P. camtschaticus*, as in other brachyuran and anomuran species [43,50,51], indicates a high degree of evolutionary conservation of this brain area in crustaceans [30,31,33,46,52].

In the deutocerebrum serotonin-immunoreactive neurons are labeled in all the studied decapods species. Most of these are olfactory interneurons that innervate the olfactory neuropils. At the early developmental (zoea I) stage in *P. camtschaticus*, 5-HT-LIR was detected in the dorsal giant neuron (DGN) in cluster 11. Unlike that in red king crab, the 5-HT-LIR DGN in crayfish and lobsters appears during the midembryonic life [13], while in spider crabs, only at the first juvenile stage [8]. However, the 5-HT-LIR DGN was reported for adults of many crustacean species [34,36,45]. The neuritis of these serotonin-containing olfactory interneurons is involved in the formation of a glomerular structure both in adults [53] and larvae [13]. Moreover, the accessory neuropils in crayfish and lobsters have been shown to be well developed and immunoreactive to 5-HT as early as during the midembryonic life [13], but undeveloped in crab larvae [8]. In *P. camtschaticus*, AcN is identified only in adult crabs, being weakly developed and not showing 5-HT-LIR. It is typical of many other brachyurans and anomurans [30,33], and is probably related to the decrease in the size and the importance of AcN in these crustaceans [30,33,49]. Small- and medium-sized 5-HT-LIR neurons have been detected in clusters 9 and 11 in king crabs after the first metamorphosis at the glaucothoe stage. Those cell clusters contain of olfactory interneurons that are involved in the modulation of olfactory processing [36,54]. Similar 5HT-LIR neurons have been described from adult *C. clypeatus*, *P. leniusculus*, *P. clarkia*, *S. olivacea*, *C. destructor*, *L. vannamei*, and *S. serrate* [30,42,43,44]. In cluster 10, 5-HT-LIR globular projection neurons are identified only in adult red king crabs. Taking into account the fact that the projection neurons of cluster 10 are involved in the transmission of olfactory signals from the first-order processing site, from the olfactory glomeruli in the olfactory neuropils, to the second-order center in the lateral protocerebrum [55], one can assume that 5-HT-LIR may be associated with the integration of olfactory and visual stimuli, as previously described from other decapods [36].

In the tritocerebrum in *P. camtschaticus*, 5-HT-LIR has been identified in the antenna 2 neuropil after the first metamorphosis at the glaucothoe stage. Earlier, strong 5-HT-LIR in this brain region was observed only in adult *Macrobrachium rosenbergii* [50] and *Litopenaeus vannamei* [42]. The serotonin immunoreactivity in the antenna 2 neuropil may indicate that serotonin plays an important role in mechanosensory and also chemosensory processing [11,31,56] in the larvae of *P. camtschaticus* during their transition from the pelagic to benthic lifestyle.

The VNC shows 5-HT-LIR all along its length in all crustaceans, but the number and position of 5-HT-LIR somata associated with the VNC vary between different species [19,28,42,43,45]. The fiber bundles are homologous in all crustacean species [27,32,57]. In the Decapoda, these are arranged into three distinct tracts, referred to as median, central, and lateral fiber bundles [28]. The neuron morphology in the thoracic ganglion of king crab larvae resembles that of comparable cells in larval and adult lobsters and crabs [28]. However, there is a heterochrony between species in the timing of emergence of 5-HT-LIR somata in the VNC and at various ontogeny stages.

Thus, in the developing lobster VNC, almost all the neuronal elements that are reactive in juvenile and adult animals are already immunolabeled in the midembryonic stage [20,28], while their counterparts in the larvae of king crabs and also the crab *Hyas araneus* become fully developed only later in larval life [8].

The planktonic larvae (zoea) stages exhibit the escape behavior through a complex mechanism of rapid strokes of the pleon. Such a behavior becomes possible provided that both the maxilliped and the pleon musculature, and its innervating nervous system, are well developed [15]. The presence of motor neurons in the fifth thoracic and first abdominal neuromeres ganglion that provide adaptive behaviors has been reported for decapod crustaceans [58].

The arrangement of these 5-HT-LIR neurons in king crabs is rather similar to those in other decapod species [8,28,45]. The anatomy of these cells suggests that they may be important in postural modulation by 5-HT [28]. As was experimentally shown, 5-HT, when injected into freely moving lobsters, generate stable and stereotyped postures [59]. This study suggest that posture is produced by a serotonin-triggered motor program [59].

Moreover, paired 5-HT neurons in the first abdominal ganglia were found in zoea I of red king crab. Similar neurons were identified in the first abdominal ganglia of lobsters [28]. A physiological analysis of these paired neurons showed their function as generalized “gain setters” for postural circuitries, amplifying the output of command neurons concerned with posture [60]. It was also shown that A1 neurons are linked to the phasic, fast muscle systems [61]. Furthermore, several studies revealed the roles of 5-HT in the control of motor circuits [62], and also other behavioral functions, including escape behavior, learning, and aggression [63].

### 4.3. Tyrosine Hydroxylase-like Immunoreactivity

The TH-like immunoreactive neuronal network present in the brain and in the VNC of *P. camtschaticus* was found across all decapods studied so far [29,42,43]. In the median brain, the TH-LIR neurons in cluster 6 with the soma location and innervation pattern characteristic were found in several species of decapods [21,29,42,64]. It is known that some interneurons in cluster 6 may be involved in the visual pathway associated with the eye. Our demonstration of TH-LIR in this cluster may also indicate its involvement in the modulation of visual inputs in this crab species. The medial protocerebral neuropils exhibit extensive TH-LIR. Protocerebral neuropils were shown to be the location through which visual input information passes and finally integrates with the ON, where chemosensory inputs from olfactory receptor neurons are processed [30,48].

In the larval deutocerebrum, the anti-TH labeling that we performed revealed the presence of neurons near the olfactory lobes in cluster 11. Earlier, DA-immunoreactive neurons in cluster 11 and fibers in the ON neuropils were also identified in the brains of *H. gammarus*, *Callinectes sapidus*, *Scylla olivacea*, *O. rusticus*, *P. argus*, *M. rosenbergii*, and *Litopenaeus vannamei* [29,42,43,65,66], suggesting that DA is involved in mediating the olfaction in crustaceans.

In the subesophageal ganglion, large TH-LIR neurons that may be homologous to the L-cell, described from several crustacean species, such as *H. gammarus* and *O. rusticus* [29,66], were detected at all larval stages of *P. camtschaticus*. L-cells become mature as early as in embryos and persist throughout the development [67]. The axon of the L-cell reaches the pericardial organ. Fort with coauthors [68] suggested that a branch of the L-cell in *C. sapidus* leaves the pericardial organ and is projected to the cardiac ganglion, with its function being to increase the heart rate in response to behavioral input. Furthermore, according to Robertson and Moulins [69], L-cells transmit information about foregut activity to the brain.

All studies on the decapod dopaminergic system mention TH-LIR neurites or neurite bundles which travel along the VNC interconnecting the segmental ganglia and connecting the VNC with the brain. Moreover, evidence has been provided that DA-immunoreactive fibers are present in the midline, central, and lateral fiber bundles, all of which extend through the SEG and thoracic ganglia [64]. The presence of the longitudinal and commissural catecholaminergic fiber tracts in the ventral aspects of the thoracic segments suggests that these fibers may serve to integrate these segments and modulate neurosecretory and/or motor function [65]. In the VNC of larval and adult red king crabs, TH-LIR somata are present in most segments. DA-immunoreactive neurons were also detected in the thoracic ganglia of adults in various decapod species (T1–T5) [29,65,66].

In the abdominal (pleonic) ganglia of *P. camtschaticus*, TH-LIR neurons were identified in the first abdominal (pleonic) ganglion. For some of crustacean species, DA-immunoreactive neurons were reported to be in the abdominal ganglia, with their distribution, however, being quite variable [42,64]. These cells are homologous to the anterior unpaired midline neurons described earlier from crayfish and lobsters [29]. In crayfish, these neurons project to the last abdominal ganglion and innervate the hindgut [70]. Some authors suggested that these dopamine immunoreactive neurons probably stimulate or enhance hindgut contractions and that they can be motoneurons or modulatory neurons [67].

## 5. Conclusions

We have provided a comparative description of the development of the serotonergic and dopaminergic neuronal networks in the major regions of the median brain and the VNC of larval and adult *P. camtschaticus*. These results indicate the complexity of the larval nervous system and provide new insight into the variety of roles that serotonin (5-HT) and dopamine (DA) may play in king crabs. Particularly, the high 5-HT and TH-LIR, and the dynamics of their distribution in the medial in the protocerebrum and deutocerebrum at all larval stages of development and in adult crabs, may indicate the potential role of 5-HT and DA in the modulation of visual and olfactory processing.

The presence and distribution of 5-HT and TH-LIR neuron and fiber tracts in the VNC suggests that they may serve to integrate these segments, as well as the function of the control of motor circuits that provide posture and also other behavioral functions, including adaptive escape behavior. We have shown that the serotonergic and dopaminergic neuronal networks are present long before the onset of metamorphosis, and then gradually develop. These become particularly pronounced at the first metamorphosis, which may be related to the development of the segmental appendages and maturation of motor behaviors in decapods. The appearance 5-HT-LIR in the antenna 2 neuropil after the first metamorphosis of the glaucothoe stage suggests that serotonin is involved in mechanosensory and also chemosensory processing. Additionally, the intensive development of the 5-HT and DA systems in the glaucothoe stage presumably reflects their potential physiological roles in mediating the settlement process in larvae and in their transition from the pelagic to benthic lifestyle.

## Figures and Tables

**Figure 1 biology-13-00035-f001:**
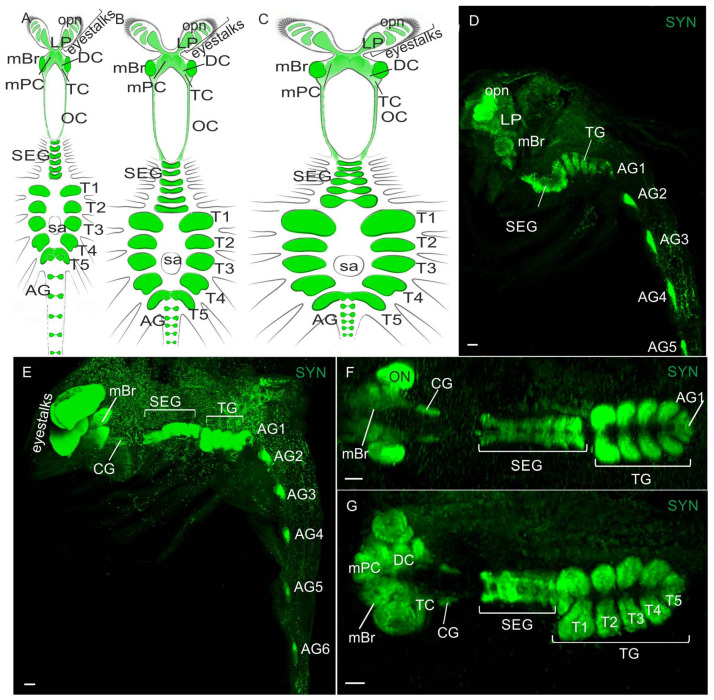
Syn-LIR in the central nervous system (CNS) at different larval stages in *P. camtschaticus,* whole mounts: (**A**–**C**) Schematic drawings of the CNS at different stages; (**A**) in zoea IV; (**B**) in glaucothoe—note the change in the neuropil size in the median brain and ventral nerve cord (VNC), especially in the 1st thoracic ganglia, and the partially reduced abdominal segments after the 1st metamorphosis; (**C**) in adult crab; (**D**,**E**) confocal image showing neuropils of median brain and VNC in the whole mounts; (**D**) in zoea I (lateral view); (**E**) in zoea IV (lateral view); (**F**,**G**) neuropils of the median brain, subesophageal, and thoracic ganglia in whole mounts: (**F**) in zoea III (dorsal view); (**G**) zoea IV (ventral view). Letter designations: eyestalks, where are localized opn—optic neuropils (lamina, medulla, lobula, and lobula plate) and LP—lateral protocerebrum; mBr—median brain; SEG—subesophageal ganglia; TG—thoracic ganglia; CG—commissural ganglion; SA—the sternal artery; AG (1–6)—abdominal ganglia; T1–T5—thoracic neuropils; mPC—median protocerebrum; DC—deutocerebrum; TC—tritocerebrum; OC—esophageal connective; ON—olfactory neuropil; LP—lateral protocerebrum; opn—optic neuropil. Green color indicates synapsin. Scale bars: 100 μm.

**Figure 2 biology-13-00035-f002:**
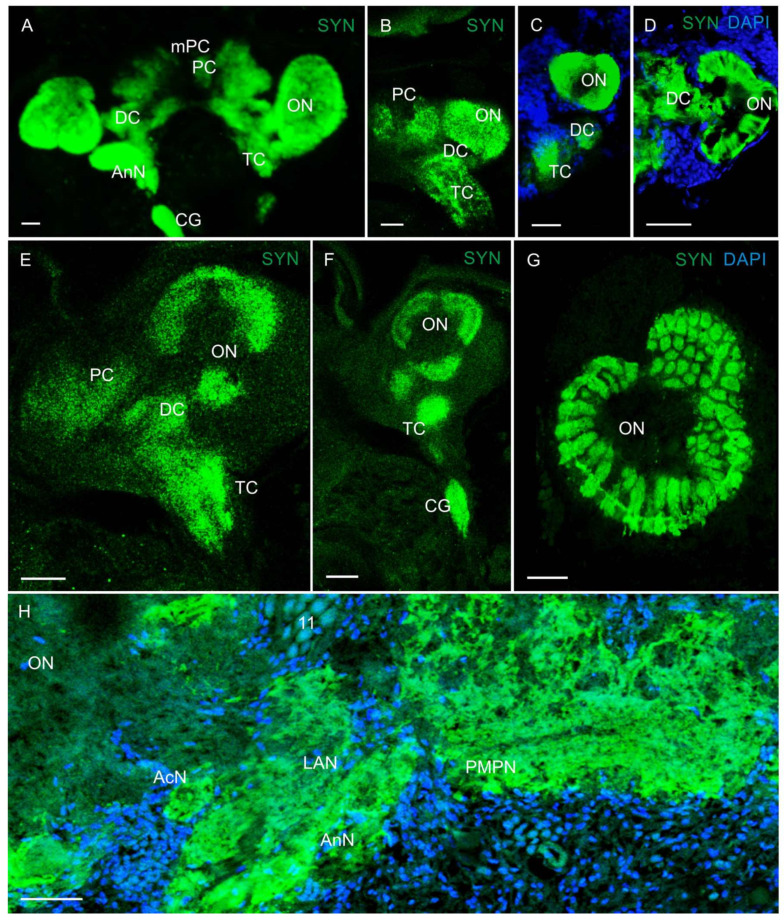
Syn-immunoreactivity of the median brain in larval and adult *P. camtschaticus*: (**A**) CLSM image of the brain in the whole mount of zoea IV (ventral view); (**B**) the brain of zoea I (ventrolateral view); (**C**) the brain of zoea I (lateral view); (**D**) cryosection through the olfactory neuropil of zoea II; (**E**) the median protocerebrum, deutocerebrum, and tritocerebrum in the whole mount of glaucothoe (dorsolateral view); (**F**) the olfactory neuropil showing a tendency to subdivide into three segments (ventrolateral view); (**G**) right olfactory neuropil in an adult crab; (**H**) cryosection through the median brain in adult crab at the level of accessory neuropils. Letter designations: mPC—median protocerebrum; PC—protocerebrum; DC—deutocerebrum; TC—tritocerebrum; CG—commissural ganglion; ON—olfactory neuropil; PMPN—posterior medial protocerebral neuropil; AcN—accessory neuropils; LAN—lateral antenna 1 neuropil; AnN—antenna II neuropil; 11—cell cluster. Green color indicates synapsin; blue color, DAPI. Scale bars: 100 μm.

**Figure 3 biology-13-00035-f003:**
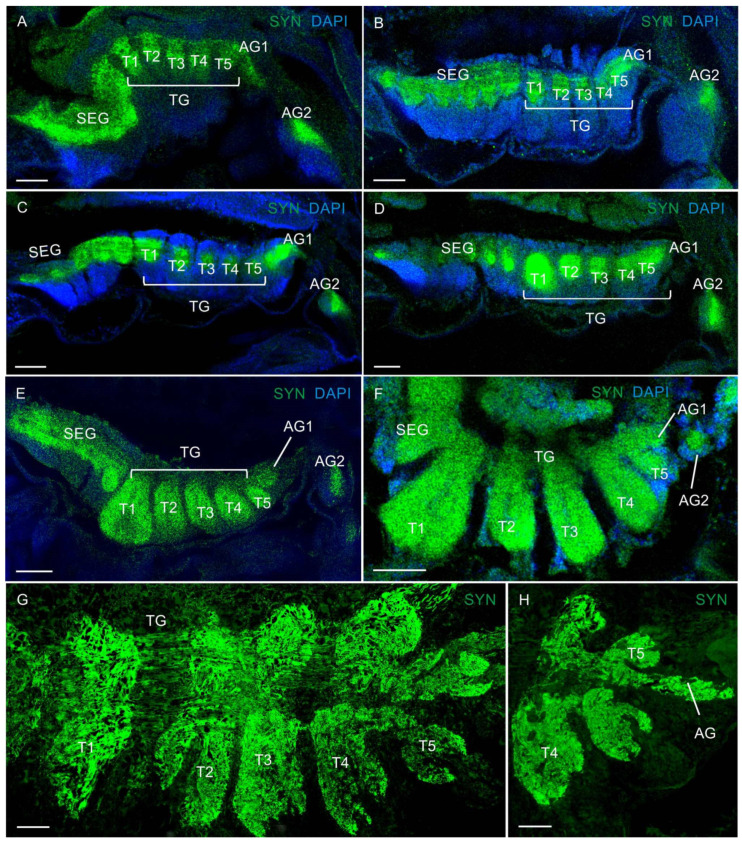
Developing neuropils and ventral nerve cord (VNC) in *P. camtschaticus*: (**A**–**H**) CLSM images showing the VNC (lateral views) in the whole mounts; (**A**) zoea I; (**B**) zoea II; (**C**) zoea III; (**D**) zoea IV; (**E**) glaucothoe; (**F**) first juvenile (crab 1) stage. (**G**) Cryosection through thoracic ganglia (4 and 5); (**H**) a fragment of the abdominal ganglia in adult crab. Letter designations: SEG—subesophageal ganglia; TG—thoracic ganglia; AG (1–6)—abdominal ganglia; T1–T5—thoracic neuropils. Green color indicates synapsin; blue color, DAPI. Scale bars: 100 μm.

**Figure 4 biology-13-00035-f004:**
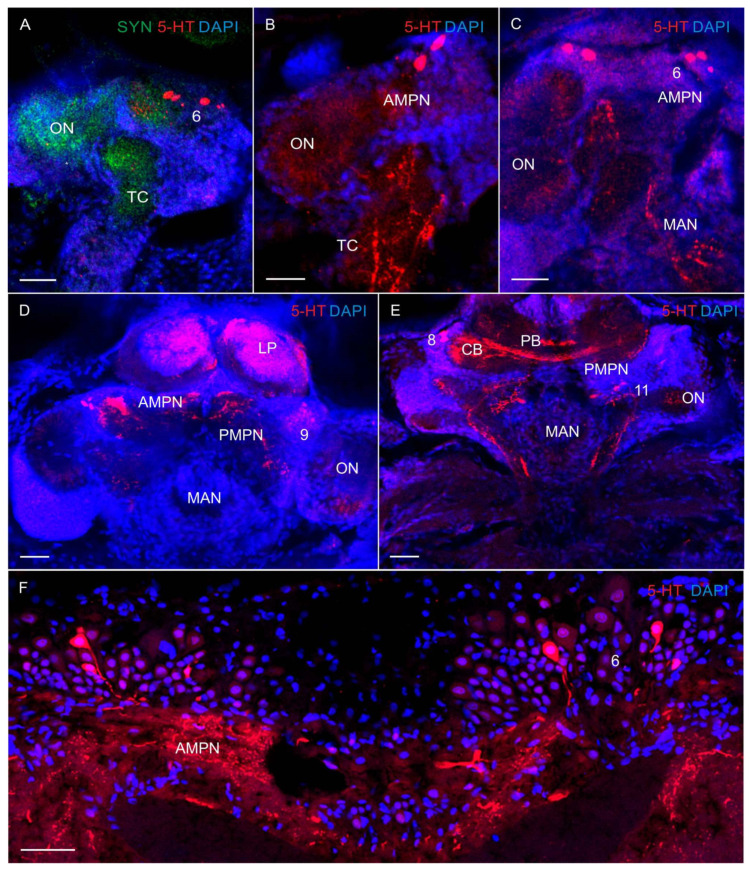
The 5-HT-like immunoreactivity in the median brain of *P. camtschaticus*: (**A**–**E**) 5-HT-LIR in the median brain whole mounts: (**A**) zoea I (lateral view); (**B**) zoea II (lateral view); (**C**) zoea III (lateral view); (**D**,**E**) glaucothoe; (**F**) adult crab. Letter designations: TC—tritocerebrum; AMPN—anterior medial protocerebral neuropil; PMPN—posterior medial protocerebral neuropil; PB—protocerebral bridge neuropil; CB—central body neuropil; MAN—median antenna 1 neuropil; ON—olfactory neuropil; LP—lateral protocerebrum; 6, 8, 9, and 11—cell clusters. Green color indicates synapsin; red color, 5-HT; blue color, DAPI. Scale bars: 100 μm.

**Figure 5 biology-13-00035-f005:**
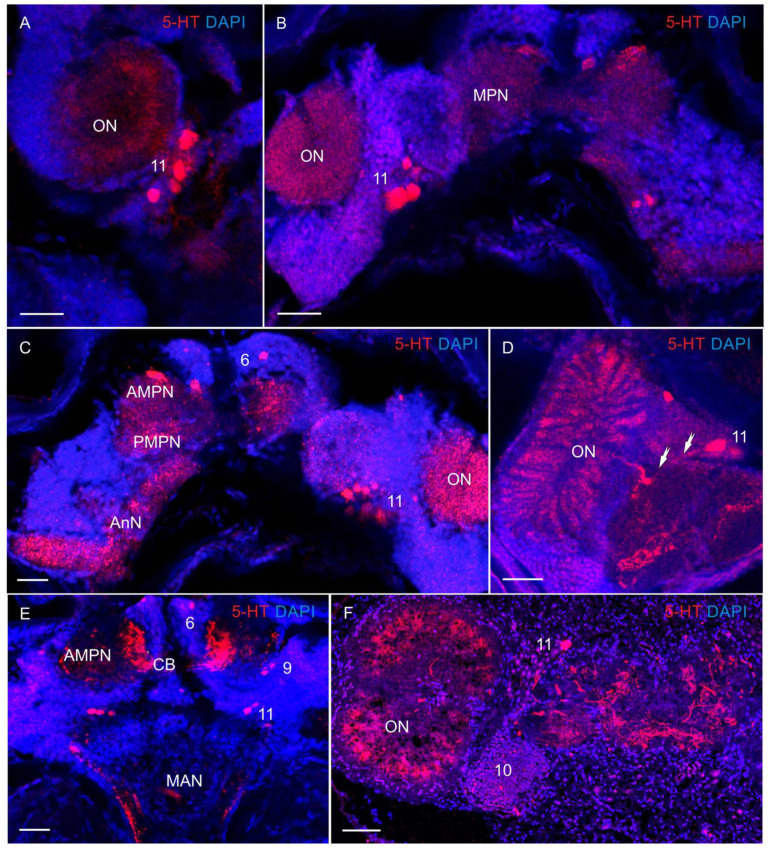
The 5-HT-like immunoreactivity in median brain of *P. camtschaticus*: (**A**) 5-HT-LIR zoea II; (**B**,**C**), glaucothoe; (**D**) first juvenile (crab 1) stage; arrows indicate projections of olfactory neurons; (**E**) 5-HT-LIR in the deutocerebrum and tritocerebrum of glaucothoe; (**F**) highly intense 5-HT-LIR labeling in the AMPN and in the glomeruli of the ON of adult crab, cryosection. Letter designations: MPNs—medial protocerebral neuropils; AMPN—anterior medial protocerebral neuropil; PMPN—posterior medial protocerebral neuropil; CB—central body neuropil; MAN—median antenna 1 neuropil; ON—olfactory neuropil; AnN—antenna II neuropil; 6, 9, 10, and 11—cell clusters. Red color indicates 5-HT; blue color, DAPI. Scale bars: 100 μm.

**Figure 6 biology-13-00035-f006:**
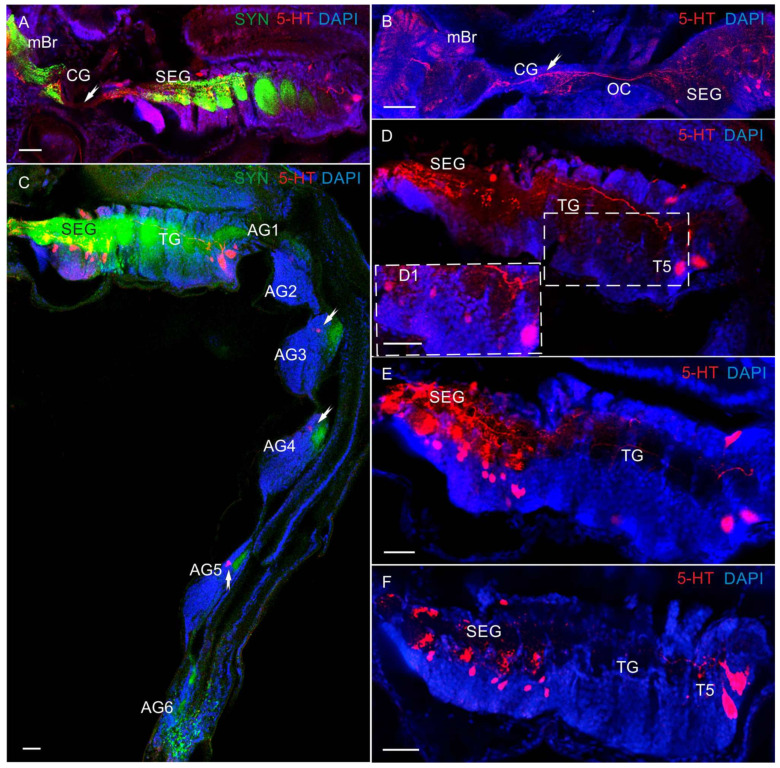
The 5-HT-like immunoreactivity in ventral nerve cord (VNC) of *P. camtschaticus*, whole mounts: (**A**) 5-HT-LIR and Syn-LIR in the commissural ganglion, subesophageal ganglia, and thoracic ganglia of zoea I; (**B**) highly intense 5-HT-immunoreactive labeling of fibers in circumesophageal connective and commissural ganglion of glaucothoe; (**C**) VNC (ventral view) of zoea II (the arrows are indicated 5-HT-LIR neurons in the AG); (**D**,**E**) 5-HT-LIR neurons and fibers in the SEG and TG of zoea II (arrows indicate low 5-HT-LIR in the neurons in neuromeres 1–3); (**F**) zoea III. Letter designations: mBr—median brain; SEG—subesophageal ganglia; TG—thoracic ganglia; CG—commissural ganglion; OC—esophageal connective; AG (1–6)—abdominal ganglia; TG—thoracic ganglia. Green color indicates synapsin; red color, 5-HT; blue color, DAPI. Scale bars: 100 μm.

**Figure 7 biology-13-00035-f007:**
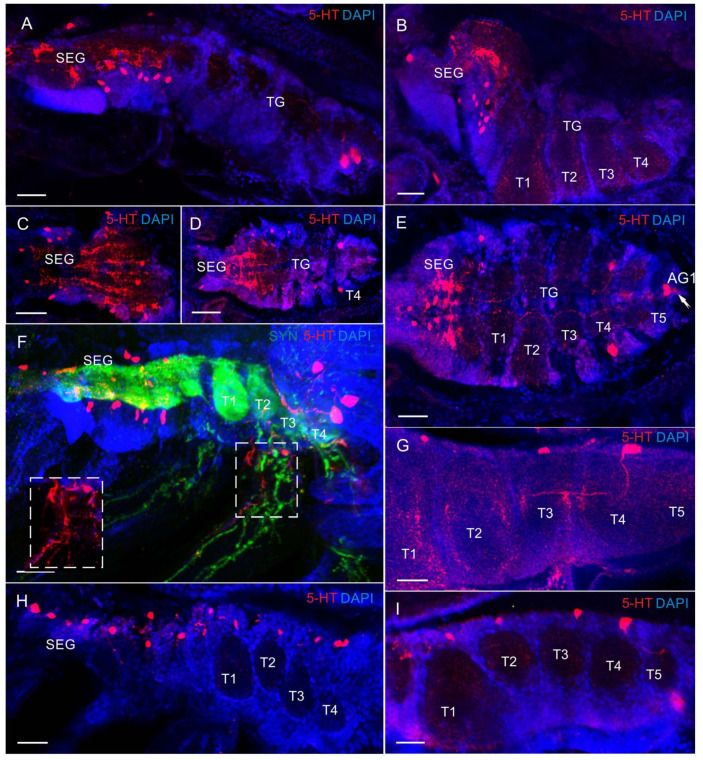
The 5-HT-like immunoreactivity in in ventral nerve cord (VNC) of *P. camtschaticus*, whole mounts: (**A**–**E**) 5-HT-LIR neurons and fibers of the VNC (**A**) in zoea IV; (**B**) glaucothoe; (**C**–**E**) first juvenile (crab 1) stage; (**E**) arrows indicate 5-HT-LIR in neuron of the 1st abdominal ganglion; (**F**) 5-HT-LIR and Syn-LIR in zoea II; (**G**) 5-HT-LIR neurons in the subesophageal ganglia and thoracic ganglia in first juvenile stage; (**H**) zoea III; (**I**) zoea IV. Letter designations: SEG—subesophageal ganglia; TG—thoracic ganglia; T1–T5—thoracic neuropils. Green color indicates synapsin; red color, 5-HT; blue color, DAPI. Scale bars: 100 μm.

**Figure 8 biology-13-00035-f008:**
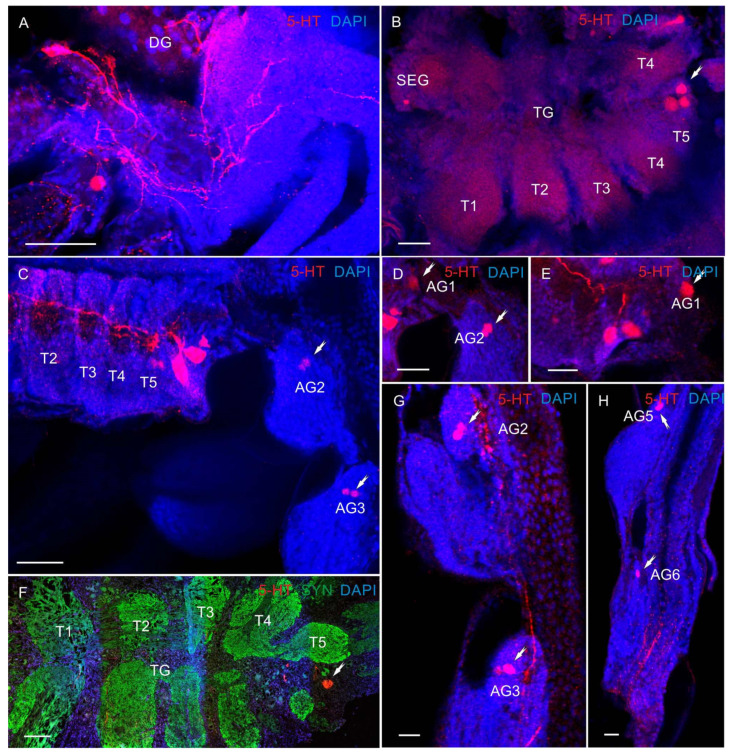
The 5-HT-like immunoreactivity in ventral nerve cord (VNC) of *P. camtschaticus*, whole mounts: (**A**) 5-HT-LIR neurons in the second thoracic neuromere of zoea II; (**B**) 5-HT-LIR neurons in the VNC of first juvenile stage; (**C**–**E**); zoea II; (**F**) cryosection through thoracic neuromere in adult crab; (**G**,**H**) abdominal ganglia in zoea III; (**B**–**H**) the arrows indicate heйpohы b AG (1–6). Letter designations: SEG—subesophageal ganglia; DG –digestive gland; TG—thoracic ganglia; T1–T5—thoracic neuropils; AG (1–6)—abdominal ganglia. Green color indicates synapsin; red color, 5-HT; blue color, DAPI. Scale bars: 100 μm.

**Figure 9 biology-13-00035-f009:**
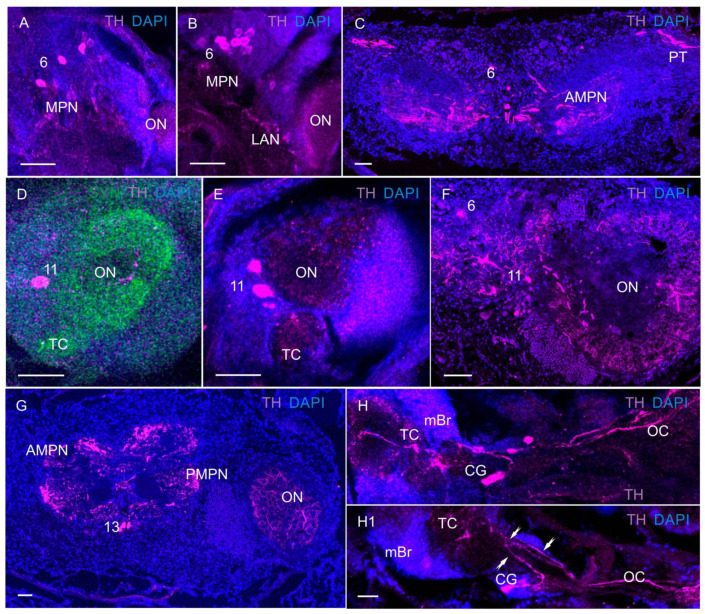
TH-like immunoreactivity in median brain at different development stages in *P. camtschaticus*, whole mounts: (**A**) TH-LIR neurons in the median protocerebrum of zoea I; (**B**) zoea IV; (**C**) adult crab; (**D**) TH-LIR neurons and fibers in deutocerebrum of zoea I; (**E**) zoea III; (**F**–**G**) adult crab; (**H**,**H1**) TH-LIR in the commissural ganglion of glaucothoe; arrows indicate that the L-cell appeared to bifurcate, sending one process towards the median brain and the other towards the VNC. Letter designations: mBr—median brain; MPN—medial protocerebral neuropils; ON—olfactory neuropil; CG—commissural ganglion; AMPN—anterior medial protocerebral neuropil; PMPN—posterior medial protocerebral neuropil; OC—esophageal connective; 6, 11, and 13—cell clusters. Green color indicates synapsin; magenta color, TH; blue color, DAPI. Scale bars: 100 μm.

**Figure 10 biology-13-00035-f010:**
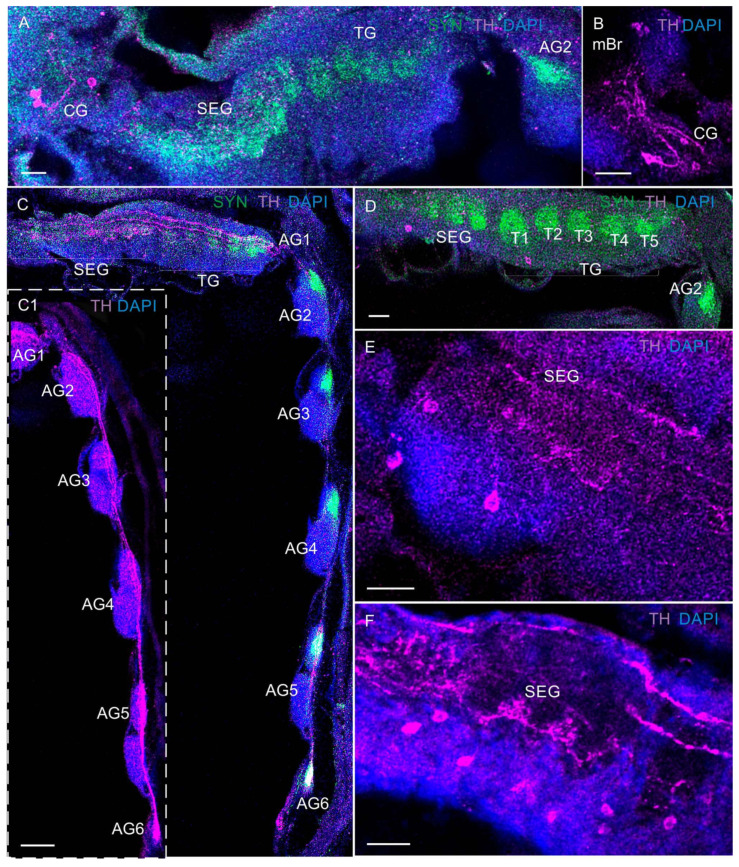
TH-like immunoreactivity and Syn-LIR in ventral nerve cord (VNC) of *P. camtschaticus*, whole mounts: (**A**) TH-LIR in the commissural ganglion and VNC of zoea II; (**B**) zoea IV; (**C**,**C1**) TH-LIR fibers projecting through the VNC in zoea III; (**D**) TH-LIR in the thoracic ganglion of zoea II; (**E**,**F**) TH-LIR in the subesophageal ganglion; (**E**) zoea I; (**F**) glaucothoe. Letter designations: mBr—median brain; CG—commissural ganglion; SEG—subesophageal ganglia; TG—thoracic ganglia; T1–T5—thoracic neuropils; AG (1–6)—abdominal ganglia. Green color indicates synapsin; magenta color, TH; blue color, DAPI. Scale bars: 100 μm.

**Figure 11 biology-13-00035-f011:**
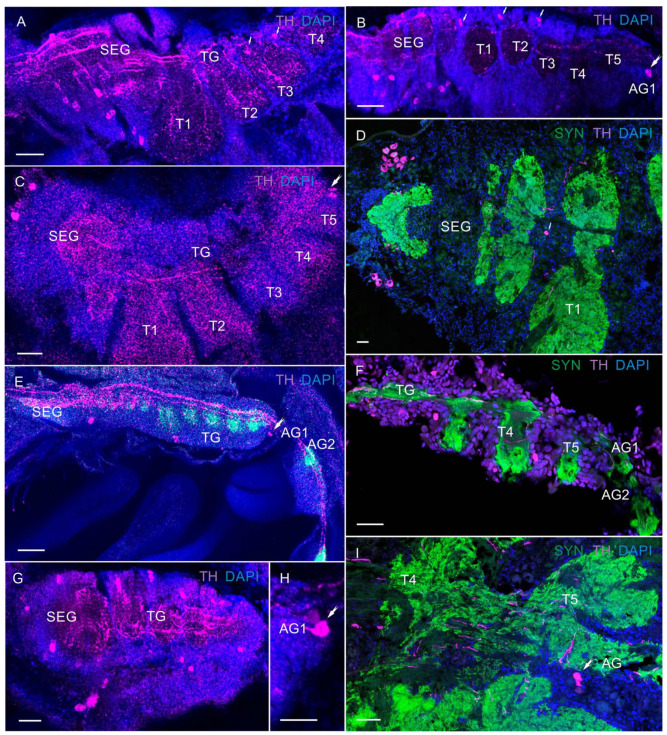
TH-like immunoreactivity and Syn-LIR in ventral nerve cord: (**A**) TH-LIR in subesophageal and thoracic ganglion of the first juvenile stage; (**B**) zoea IV; (**A**,**B**,**D**) little arrows indicate the medial neurons b thoracic ganglia; (**C**) first juvenile stage (**D**) and adult crab; (**E**) TH-LIR in the thoracic and abdominal ganglia of zoea III; (**F**) zoea IV, cryosection; (**G**,**H**) glaucothoe; (**I**) adult crab; (**B**,**C**,**E**,**H**,**I**), big arrows indicate the neurons b AG1. Letter designations: SEG—subesophageal ganglia; TG—thoracic ganglia; T1–T5—thoracic neuropils; AG (1–6)—abdominal ganglia. Green color indicates synapsin; magenta color, TH; blue color, DAPI. Scale bars: 100 μm.

**Table 1 biology-13-00035-t001:** Brief characteristics of ontogeny stages of red king crab *P. camtschaticus*.

Stages	Duration	Organs of Locomotion	Habit of Life	Carapace Length (±SD) mm
zoea I 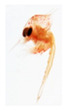	8–9 days	Exopoditesof maxillipeds I–II	Planktonic	1.39 ± 0.08
zoea II 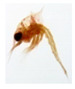	7–8 days	Exopoditesof maxillipeds I–III	Planktonic	1.43 ± 0.07
zoea III 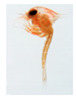	9–11 days	Exopoditesof maxillipeds I–III	Planktonic	1.83 ± 0.1
zoea IV 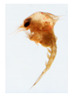	10–13 days	Exopoditesof maxillipeds I–III	Planktonic	2.07 ± 0.12
glaucothoe 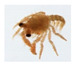	14–17 days	Pleopods	Plankton–benthic	1.85 ± 0.13
first juvenile 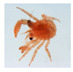	18–20 days	Pereiopods II–IV	Benthic	1.91 ± 0.1

## Data Availability

The authors confirm that the data supporting the findings of this study are available within the article. In addition, the raw data that support the findings of this study are available on request from the corresponding author.

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
