# Peer review of "Development of Serotonergic and Dopaminergic Neuronal Networks of the Central Nervous System in King Crab, Paralithodes camtschaticus"

_biology, 2024, doi:10.3390/biology13010035_

Round 1

Reviewer 1 Report

Comments and Suggestions for Authors

The manuscript provides the description of the pattern of central nervous system development in Paralithodes camtschaticus, which offers a map of the neural architecture and distribution of cells producing serotonin and dopamine in the medial brain and ventral nerve cord  of the larval nervous system. The results could be important for better understanding of the neural mechanism of behavior at different developmental stages of the red king crab. The work is properly designed, and the results are clearly presented. I recommend it for publication in Biology. My only suggestion is that it would be better if the authors can add diagram which can summerize the results.

Author Response

The manuscript provides the description of the pattern of central nervous system development in Paralithodes camtschaticus, which offers a map of the neural architecture and distribution of cells producing serotonin and dopamine in the medial brain and ventral nerve cord  of the larval nervous system. The results could be important for better understanding of the neural mechanism of behavior at different developmental stages of the red king crab. The work is properly designed, and the results are clearly presented. I recommend it for publication in Biology. My only suggestion is that it would be better if the authors can add diagram which can summerize the results.

Reply

Dear Reviewer 1.

We have added a table of larval stages for the convenience of presenting the material and Figure 1 shows a schematic picture of the structure of the central nervous system at different stages of crab development

Reviewer 2 Report

Comments and Suggestions for Authors

General Comments

The manuscript 2758630 submitted by V. Dyachuk describes the presence of serotonergic and dopaminergic neural structures in Paralithodes camtschaticus. The work is quite extensive as the analyses are conducted across different developmental stages. The quality of the analyses and images appears to be high and the inferences made are valid. The fact that this research work adds to studies conducted on different species previously, may appear as lack of novelty. It is, however, not so, because adding another species increase knowledge and allows for further comparison, confirmation of commonalities as well as differences between taxonomic groups and species. In fact, this is what the authors do in their discussion and which I find worthwhile indeed. Nevertheless, the authors may increase their effort in pointing out the practical implications (e.g., as in L. 678-780), which are rather vague and too concise as compared to the extensive descriptive aspects of the work. One general problem of the manuscript is that it provides very detailed descriptions that are, because of the different developmental stages, often redundant. The meticulous descriptions are very honorable and validate the amount of work invested into this study. However, the reading becomes very laborious and the differences and consistencies with other taxa are rather difficult to gather. The point I am trying to make is the following: If one seeks for detailed information the text is certainly excellent, but a quick read is practically impossible. There may be two solutions to this (depending on the editor’s policy on word count): a) Commence or finish each section with a very condensed resumé (sort of an extended subheading) preceded or followed with the detailed descriptions; b) outsource some of the more detailed information into supplementary data. I am aware of the latter suggestion being difficult to accept and/or to realize, but the first solution will make the text even longer. The authors should keep in mind that they wish to be read and there might be only a handful of readers that want to delve into the very details of serotonergic and dopaminergic neurons in P. camtschaticus. If the essence of the work is not better carved out, I doubt that many readers will indeed read the article to the end and this would be an even greater waste than making the reading more concise and pertinent. It may be argued that this is done with the conclusions, but then again with 23 lines these are probably too concise. Perhaps a compromise could be found.

Detailed comments

Abstract: If, in the abstract a term is used only once (or twice), no need for introducing an abbreviation, which should generally be avoided in the abstract. The second part of the abstract is not clear and contains practically no results. Please rewrite.

Introduction:

L. 36: “… despite their high diversity.”

L. 40: remove “target”

L. 47: specify what you understand by “rich behavioural repertoire” Why would this differ from other crabs and what is the relevance for the study (5-HT, DA act on behaviour).

L. 53: “… changes in their nervous system.”

L. 62-63: Please rephrase! “maturation of neurotransmitters” ???

L. 63: The authors point to the role of neurotransmitters in larval development. This is exactly what I have referred to in my general comments. The work is highly descriptive, but, eventually, it does not clearly inform about functional inferences that can be made from the observations.

L. 71: “… distributed in developing …”

L. 95: “… by Anger and Nair …”

L. 95-96: What was the salinity and oxygenation; how much food was supplied?

L. 97: What do the authors understand by “stable larval development”?

L. 102: This is confusing! The authors keep the larvae at 3°C, but declinate the larval development for 7-8°C.

L. 111-112: How do the authors define ‘adulthood’? What was the size and the volume of the tanks and how many animals were kept in one tank?

L. 114: What is natural photoperiod? This depends on the tie of the year and is highly variable. Please define precisely. How much were the animals fed?

L. 117: Have the authors checked how MgCL2 affects the neurotransmitters?

L. 123-125: Thermo Fisher is once located in the US and once in Germany. Best would be to refer to the headquarters.

L. 130: As far as I am aware of, I do not see any means ± SD in the results section and surely no statistics on the measures. Why?

L. 135: Perhaps specify that TH is the enzyme that catalyzes tyrosine into DOPA, the precursor of DA.

L. 139: Specify the supplier of normal donkey serum!

L. 142: Once you have introduced a supplier, it is not necessary to recall its location every time you refer to this supplier

L. 147: Better treat Invitrogen as a company of its own.

L.160-161: “Abs” is an abbreviation not introduced and used only twice. If you want to use it, first introduce it and then use it consequently.

L. 161-172 and ff.: No need to mention locations for suppliers already introduced.

L. 195: Which company supplies SYNORF1 or 3C11?

L. 212: It might be useful to provide a table with the abbreviations in an alphabetical order. There might be a conflict between on – optic neuropil and ON – olfactory neuropil. Perhaps the optic neuropil can be designated differently? It does appear less often than ON.

In the following results part, many abbreviations are introduced and the whole designation is used again with the abbreviation in parentheses. The point of introducing an abbreviation is to replace the whole term by this abbreviation. The authors should do this consistently throughout the manuscript, or if the prefer, not use the abbreviations (whichever way, but not both).

Results:

Fig. 1: It seems to me that not all structures are consistently identified. In fig. 1 E and F, what are the two large ganglia on the left-hand side? LP ON? There is also no CG in fig. 1 E. “sa” seems to missing from the letter designations. And: I see no blue color corresponding to DAPI. Generally: check all figures for completeness of labelling of structures and completeness of legends.

L. 237: The abbreviation CNS has not been introduced in the figure legend. Do like in fig. 10 consistently in all figures! But only if abbreviation is used more than once (like in fig. 10 and 11)!

L. 249: The measures in the results section are provided as approximately hundreds of micrometers (~ 280 μm etc.). However, the material and methods section specifies a number of 10 animals with a mean and a standard deviation. This should be used in the results section instead of  ~. By doing so, statistically significant differences could be calculated and one could appreciate if the increase in size is significant and meaningful (even if the trends are clear). In any case, it would be scientifically more appropriate than an approximate size measure.

L. 265: To my understanding each figure legend has to provide stand alone information. Consequently, each abbreviation has to be introduced. It should not be necessary to seek for the abbreviation in the text. A solution to this could be a list or, even better, a table and then the authors could refer to this. But better just introduce occasionally within the figure legend where necessary as you already do this for all the other terminology.

L. 283: This is an example of what I referred to above. The abbreviation for suboesophageal ganglion was already introduced. The designation in its full length is no longer needed.

L. 289: Same thing: TG! And so on throughout the manuscript.

L. 304: “… 5th thoracic ganglion …”

Fig. 5D: Should the massively 5-HT stained structures in the upper part of the image not be labelled?

Fig. 6: I think that insert D1 does not provide more information. Can be removed. For fig. 6 B and C there are arrows; what do they indicate (“Arrows indicate …”). This information should be included in the legend. How comes that there is no 5-HT-LIR neuron in AG2.

L. 385-386: “… low 5-HT-LIR intensity were detected …” perhaps indicate by arrows.

L. 390: “… in each of the pereion segments …”

Fig. 7: What does the arrow indicate? Please specify.

Fig. 8: What does the arrow indicate? Please specify.

Fig. 9H1: What does the arrow indicate? Please specify.

Fig. 10: What does the arrow indicate? Please specify.

Fig. 11B: The arrows do not have the same size. And: “Arrows indicate …”

Discussion:

L. 488: “…, this study …”

L. 439: The authors emphasize the olfactory input in the development of the ON, which they have identified to be important. However, I am surprised that the ganglions of the optic nerve and the optic input appear completely underrepresented. In fact, the optic neuropil and the lateral protocerebrum are present in fig. 1 A-D, but they do not appear in the following, despite their importance and the importance of serotoninergic innervation in the eyestalk. There is one sole reference to the visual input (49 in l. 536). It, therefore, seems that the importance of the optic input throughout development has been widely neglected by the authors. Have the eyestalks not been included into the analyses? I feel that there is something missing, which at least needs to be discussed in this section. Perhaps extend following l. 536. The visual pathway is also mentioned in l. 618. Perhaps regroup these observations.

L. 504: Generally speaking: on how many repeated observations are the statements made? To my point of view, this is one of the major problems in histology, that a conclusion may be draw from one observation. Given that ten animals have been analyzed, it would be good to have some information about the number of observations mad that support the conclusions.

L. 546: remove “deuterocerebral”

L. 562: “Those cell clusters is a localization of …” I do not understand. Please rephrase.

L. 589: This is entirely descriptive. Do the authors have any explanation for this difference? Can they come up with a functional hypothesis for why P. camtschaticus and Hyas araneus develop differently? Perhaps this is what the authors try to elaborate in the following (l.591-597, but I don’t see how it would explain the differences between lobster, King crab and Spider crab.

L. 603-604: Only one study cited, so it must read “This study suggests that  …”

L. 617: Some Cyrillic language is present in this line. Translate or remove!

Conclusions:

The first two sentences of the conclusion section emphasize that new insights are provided. Well, this is somewhat relative as descriptions of the serotonergic and dopaminergic neurons in other species exist. That they control motor circuits and behavior is not new.

Again, the processing of olfactory as well as mechanosensory and chemosensory input by 5-HT and DA is worthwhile to be outlined, but the optic input is equally important. This, however, is not mentioned. Would the authors take the position in saying the development of the ON indicates that the olfactory input is more important, whilst not having analyzed the optic neuropils?

L. 673: Use abbreviations consistently! Here the authors change between 5-HT and DA, the use “serotonin or dopamine” and again “5-HT and DA”. Please take care to check the entire manuscript for such inconsistencies.

L. 647: I think the reference to Barnacles is not really pertinent, as these undergo a more profound metamorphosis and attach, which is not the case with King crabs.

L. 678-680: Can the authors become a little more specific on the electrophysical and pharmacological approaches?

Comments on the Quality of English Language

The English is generally very good with some minor corrections to be made here and there.

Author Response

General Comments

The manuscript 2758630 submitted by V. Dyachuk describes the presence of serotonergic and dopaminergic neural structures in Paralithodes camtschaticus. The work is quite extensive as the analyses are conducted across different developmental stages. The quality of the analyses and images appears to be high and the inferences made are valid. The fact that this research work adds to studies conducted on different species previously, may appear as lack of novelty. It is, however, not so, because adding another species increase knowledge and allows for further comparison, confirmation of commonalities as well as differences between taxonomic groups and species. In fact, this is what the authors do in their discussion and which I find worthwhile indeed. Nevertheless, the authors may increase their effort in pointing out the practical implications (e.g., as in L. 678-780), which are rather vague and too concise as compared to the extensive descriptive aspects of the work. One general problem of the manuscript is that it provides very detailed descriptions that are, because of the different developmental stages, often redundant. The meticulous descriptions are very honorable and validate the amount of work invested into this study. However, the reading becomes very laborious and the differences and consistencies with other taxa are rather difficult to gather. The point I am trying to make is the following: If one seeks for detailed information the text is certainly excellent, but a quick read is practically impossible. There may be two solutions to this (depending on the editor’s policy on word count): a) Commence or finish each section with a very condensed resumé (sort of an extended subheading) preceded or followed with the detailed descriptions; b) outsource some of the more detailed information into supplementary data. I am aware of the latter suggestion being difficult to accept and/or to realize, but the first solution will make the text even longer. The authors should keep in mind that they wish to be read and there might be only a handful of readers that want to delve into the very details of serotonergic and dopaminergic neurons in P. camtschaticus. If the essence of the work is not better carved out, I doubt that many readers will indeed read the article to the end and this would be an even greater waste than making the reading more concise and pertinent. It may be argued that this is done with the conclusions, but then again with 23 lines these are probably too concise. Perhaps a compromise could be found.

Reply

Thank you for your valuable comments

Detailed comments

If, in the abstract a term is used only once (or twice), no need for introducing an abbreviation, which should generally be avoided in the abstract. The second part of the abstract is not clear and contains practically no results. Please rewrite.

Reply

Abstract: The second part of the abstract was rewritten

Introduction:

  1. 36: “… despite their high diversity.”

Reply

  1. 36: “… account their high diversity” changed to despite their high diversity

  1. 40: remove “target”

Reply

removed

  1. 47: specify what you understand by “rich behavioural repertoire” Why would this differ from other crabs and what is the relevance for the study (5-HT, DA act on behaviour).

Reply

  1. 47: In decapod crustaceans, zoea larvae live in the water column and control the vertical parameters of the habitat and can regulate environmental factors: illumination, hydrostatic pressure, tidal currents, temperature, salinity and food concentration [5]. Changes in behavior are caused by living conditions. When the larvae of decapod crustaceans encounter adverse conditions, they can also respond with active avoidance behavior.

  1. 53: “… changes in their nervous system.”

Reply

  1. 53: Done

Reply

  1. 63: The authors point to the role of neurotransmitters in larval development. This is exactly what I have referred to in my general comments. The work is highly descriptive, but, eventually, it does not clearly inform about functional inferences that can be made from the observations.

Reply

  1. 63: the text has been changed: «Although advances have been made to elucidate the anatomy of anomurans’ nervous system at certain developmental stages [12, 13, 15], there remains poor understanding neurotransmitter specialization of certain types of neurons as well as development of neurotransmitter systems and their possible role in the larval development stages and adult crabs.»

  1. 71: “… distributed in developing …”

Reply

  1. 71: done

  1. 95: “… by Anger and Nair …”

Reply

  1. 95: OK

  1. 95-96: What was the salinity and oxygenation; how much food was supplied?

Reply

  1. 95-96: The larvae were kept in 400l of basin complexes, their planting density was 50 individuals per 1 liter.The larvae were kept at a constant temperature of 7-8°C, a salinity of 30–31‰, a dissolved oxygen concentration of 8.1–8.5 mg/L and under a natural light/dark cycle of 12: 12 h, and fed brine shrimp nauplii, Artemia sp. 2 раза в сутки.
  2. 97: What do the authors understand by “stable larval development”?

Reply

  1. 97: The basin complexes maintained stable conditions for development, constant water temperature, salinity, sufficient feed, low density of larvae, and the absence of predators. The larvae passed through the stages of development relatively evenly

  1. L. 102: This is confusing! The authors keep the larvae at 3°C, but declinate the larval development for 7-8°C.

Reply

  1. 102: The typo has been corrected. The water temperature during the incubation period of the eggs before hatching was gradually increased from 3 to 7 ° C. During the larval period of development, the larvae were kept at the same temperature of 7-8 ° C, as well as adult crabs.

  1. 111-112: How do the authors define ‘adulthood’? What was the size and the volume of the tanks and how many animals were kept in one tank?

Reply

  1. 111-112: Adult crabs were kept in aquariums with a volume of 40 liters, 1 animal in each aquarium. The average size of the male shell is 15-16 cm, with a leg span of 1 m. The average weight of males is 1.5–2.5 kg.
  2. 114: What is natural photoperiod? This depends on the tie of the year and is highly variable. Please define precisely. How much were the animals fed? Л. 114:

Reply

  1. 114: The photoperiod is measured in hours and is usually expressed as the ratio of daylight to dark time. The animals were kept under natural light light/dark cycle of 12: 12 h, the experiment was conducted in March-April 2022. Adult crabs were fed fresh blue mussels (Mytilus edulis) once a day

  1. 117: Have the authors checked how MgCL2affects the neurotransmitters?

Reply

Anesthesia serves as an effective method to mitigate the stress response in animals. Currently, one of the most effective and safe anesthetics for some crustaceans, however, it is known that it can increase in inhibitory neurotransmitters. However, the concentrations used by us cause a weak effect and did not cause changes in the level of 5-HT, TN and CHAT in the nerve ganglia in Kamchatka compared with the control (low-temperature exposure).

  1. 123-125: Thermo Fisher is once located in the US and once in Germany. Best would be to refer to the headquarters.

Reply

OK

  1. 130: As far as I am aware of, I do not see any means ± SD in the results section and surely no statistics on the measures. Why?

Reply

The dimensional data were given in Table 1

  1. 135: Perhaps specify that TH is the enzyme that catalyzes tyrosine into DOPA, the precursor of DA.

Reply

  1. 135: Added text « Immunostaining was used to determine the distribution of serotonin or tyrosine hydroxylase (TH), the enzyme that catalyzes tyrosine into DOPA, the precursor of DA.»

  1. 139: Specify the supplier of normal donkey serum!

Reply

  1. 139: OK «  normal donkey serum (NDS) (Jackson ImmunoResearch, Cambridge House, St. Thomas Place, UK)» 

  1. 142: Once you have introduced a supplier, it is not necessary to recall its location every time you refer to this supplier

Reply

  1. 142: OK

  1. 147: Better treat Invitrogen as a company of its own.

Reply

  1. 147: Done

L.160-161: “Abs” is an abbreviation not introduced and used only twice. If you want to use it, first introduce it and then use it consequently.

Reply

L.160-161:   «  polyclonal Abs  »    changed to   «  polyclonal antibody  »

  1. 161-172: and ff.: No need to mention locations for suppliers already introduced.

Reply

  1. 161-172: Deleted
  2. 195: Which company supplies SYNORF1 or 3C11?

Reply

  1. 195: 3C11 (anti SYNORF1) in mouse, Developmental Studies Hybridoma Bank (DSHB Cat# 3C11, RRID: AB528479)
  2. 212: It might be useful to provide a table with the abbreviations in an alphabetical order. There might be a conflict between on – optic neuropil and ON – olfactory neuropil. Perhaps the optic neuropil can be designated differently? It does appear less often than ON.

Reply

  1. 212: « on – optic neuropil»   changed to  «  oрn – optic neuropil»  

In the following results part, many abbreviations are introduced and the whole designation is used again with the abbreviation in parentheses. The point of introducing an abbreviation is to replace the whole term by this abbreviation. The authors should do this consistently throughout the manuscript, or if the prefer, not use the abbreviations (whichever way, but not both).

Results:

Fig. 1: It seems to me that not all structures are consistently identified. In fig. 1 E and F, what are the two large ganglia on the left-hand side? LP ON? There is also no CG in fig. 1 E. “sa” seems to missing from the letter designations. And: I see no blue color corresponding to DAPI. Generally: check all figures for completeness of labelling of structures and completeness of legends.

Reply

In Fig. 1, eyestalks designations have been added, in which on - optic neuropils (lamina, medulla, lobula, and lobula plate) and LP - lateral protocerebrum, SA - the sternal artery are localized, CG - commissural ganglion has been designated in Fig. 1 E and DAPI has been removed from the list of symbols to Figure 1. In fig. 1 E in eyestalks, individual neuropiles are not identified due to high autofluorescence.

  1. 237: The abbreviation CNS has not been introduced in the figure legend. Do like in fig. 10 consistently in all figures! But only if abbreviation is used more than once (like in fig. 10 and 11)!

Reply

  1. 237: The abbreviation CNS have added in the figure legend. Changes have been made to the captions to Fig. 1

  1. 249: The measures in the results section are provided as approximately hundreds of micrometers (~ 280 μm etc.). However, the material and methods section specifies a number of 10 animals with a mean and a standard deviation. This should be used in the results section instead of  ~. By doing so, statistically significant differences could be calculated and one could appreciate if the increase in size is significant and meaningful (even if the trends are clear). In any case, it would be scientifically more appropriate than an approximate size measure.

Reply

Yes, we agree with this remark, but the purpose of this pioneer is to detect the localization of serotonin and dopamine neurons in the crab central nervous system. Of course, further work will be related to their quantitative analysis.

  1. 265: To my understanding each figure legend has to provide stand alone information. Consequently, each abbreviation has to be introduced. It should not be necessary to seek for the abbreviation in the text. A solution to this could be a list or, even better, a table and then the authors could refer to this. But better just introduce occasionally within the figure legend where necessary as you already do this for all the other terminology.

Reply

Thank you for the recommendation. We fixed it

  1. 283: This is an example of what I referred to above. The abbreviation for suboesophageal ganglion was already introduced. The designation in its full length is no longer needed. Л. 283:.

Reply

  1. 283: suboesophageal ganglion (SEG) changed to SEG

  1. 289: Same thing: TG! And so on throughout the manuscript.

Reply

  1. 289: thoracic ganglion changed to TG

  1. 304: “… 5ththoracic ganglion …”

Reply

  1. 304: «…5st thoracic ganglion » changed to «  5th thoracic ganglion » …”

Fig. 5D: Should the massively 5-HT stained structures in the upper part of the image not be labelled?

Reply

No. In Fig. The olfactory neuropile and olfactory interneurons are shown

Fig. 6: I think that insert D1 does not provide more information. Can be removed. For fig. 6 B and C there are arrows; what do they indicate (“Arrows indicate …”). This information should be included in the legend. How comes that there is no 5-HT-LIR neuron in AG2?

Reply

Figure 6: Insert D1 has been removed. In Fig. Arrows 6C point to 5-HT-IR neurons in AG. Added explanations in the legend. Larvae found in crabs have an increased frequency of VNC, cells in AG1-6 are not always possible to detect in one area, therefore, the number of 5-HT-LIR neurons in AG2 was additionally indicated in Fig. 8C, D and G.

  1. 385-386: “… low 5-HT-LIR intensity were detected …” perhaps indicate by arrows.

Reply

  1. 385-386: 6D arrows marked neurons with low 5-HT-IR in thoracic neuromeres 1-3. Added explanations in the legend.
  2. 390: “… in each of the pereion segments …”

Reply

  1. 390: « in each of pereion segments »   заменили  на   «   in each of the pereion segments  »

Fig. 7: What does the arrow indicate? Please specify.

Reply

На Fig. 7Е: the arrow indicate 5-HT-LIR нейрон in the 1st abdominal (pleonic) ganglion   внесены изменения в подписи к рисунку

Fig. 8: What does the arrow indicate? Please specify. Рис. 8:

Reply

На Fig. 8 B- H: the arrows indicate нейроны в AG (1–6)     внесены изменения в подписи к рисунку

Fig. 9H1: What does the arrow indicate? Please specify.

Reply

Fig. 9H1: arrow indicate, that L-cell appeared to bifurcate, sending one process towards the median brain and the other towards the VNC.

Fig. 10: What does the arrow indicate? Please specify.

Reply

Fig. 10: Нет стрелок

Fig. 11B: The arrows do not have the same size. And: “Arrows indicate …”

Reply

Fig. 11B: little arrows indicate medial нейроны в thoracic ganglia, big arrows indicate нейроны в AG1.

Discussion:

  1. 488: “…, this study …”

Reply

  1. 488: “… « a study» replace to , «    this study  »…”          

  1. 439: The authors emphasize the olfactory input in the development of the ON, which they have identified to be important. However, I am surprised that the ganglions of the optic nerve and the optic input appear completely underrepresented. In fact, the optic neuropil and the lateral protocerebrum are present in fig. 1 A-D, but they do not appear in the following, despite their importance and the importance of serotoninergic innervation in the eyestalk. There is one sole reference to the visual input (49 in l. 536). It, therefore, seems that the importance of the optic input throughout development has been widely neglected by the authors. Have the eyestalks not been included into the analyses? I feel that there is something missing, which at least needs to be discussed in this section. Perhaps extend following l. 536. The visual pathway is also mentioned in l. 618. Perhaps regroup these observations.

Reply

  1. L. 439: In accordance with the reviewer's recommendation, we have added text about the role of 5-HT and DA the modulation of visual inputs in this Kamchatka crab in the discussion in section 4.2. Serotonin-Like Immunoreactivity. In this study, in the “Materials and Methods” section, we indicated that in the present study, we did not consider the neuropils located in the eyestalk, including a part of the lateral protocerebrum (hemiellipsoid body and terminal medulla). This is due to objective reasons: 1. The CLSM method we use on whole mounts does not allow us to obtain high-quality high-quality photos and show the dynamics of the appearance of 5-HT and DA in optical neuropiles, sinus gland, hemiellipsoid body and terminal medulla at different stages of larval development due to the strong autofluorescence of the eyes in larvae at the zoae1 -4 stage. Previously, in adult crabs, we studied the distribution of neurotransmitters in the eyestalks of adult mature crabs of Paralithodes camtschaticus on serial cryostat sectionsKotsyuba E, Dyachuk V. Immunocytochemical Localization of Enzymes Involved in Dopamine, Serotonin, and Acetylcholine Synthesis in the Optic Neuropils and Neuroendocrine System of Eyestalks of Paralithodes camtschaticus. Front Neuroanat. 2022 Apr 8;16:844654. doi: 10.3389/fnana.2022.844654. PMID: 35464134; PMCID: PMC9024244. Studies on serial cryostatic sections at several larval stages would add a large number of photographs and would significantly increase the volume of the article.

  1. 504: Generally speaking: on how many repeated observations are the statements made? To my point of view, this is one of the major problems in histology, that a conclusion may be draw from one observation. Given that ten animals have been analyzed, it would be good to have some information about the number of observations mad that support the conclusions.

Reply

  1. 504: Table 1 shows the main stages of development of the Kamchatka crab, which we studied. 10 larvae were selected at each stage. The neuroanatomy of their central nervous system was studied by confocal laser scanning microscopy, the larvae were stained with anti‑synapsin antibody and DAPI. There were no differences in the structure of the central nervous system on the scans within one stage. Based on the data obtained, a diagram of successive changes in the Median brain and central nerve cord has been compiled. The neuroanatomy of adult crabs has been studied previously, so 3 mature adult crabs were used for the study

  1. 546: remove “deuterocerebral”

Reply

  1. 546: “deuterocerebral” removed
  2. 562: “Those cell clusters is a localization of …” I do not understand. Please rephrase.

Reply

  1. 562: «Those cell clusters is a localization of » changed to « Those cell clusters contain olfactory interneurons that are »      . 

  1. 589: This is entirely descriptive. Do the authors have any explanation for this difference? Can they come up with a functional hypothesis for why P.camtschaticusand Hyas araneus develop differently? Perhaps this is what the authors try to elaborate in the following (l.591-597, but I don’t see how it would explain the differences between lobster, King crab and Spider crab.
    Reply

Л. 589: According to Geiselbrecht H, Melzer RR. 2013 " On the neuronal level segmental ganglia and nerves reflect different developmental plateaus of the larval body segments and tagmata; for example, segments with already well‐developed appendages possess well‐developed ganglia as well, wherever in segments without limbs or limb buds the morphogenesis of ganglia is also at a less advanced stage. Different sets of nervous system characters are thus revealed for the studied species, correlated with different types of external morphology" The data obtained on Kamchatka crabs are consistent with the well-known point of view that Kamchatka crabs are consistent with the well-known statement, ".. that the patterns of neurogenesis in the ventral ganglia of decapod crustaceans are intimately related to the development of the segmental appendages and maturation of motor behaviors ". We have studied the development of the central nervous system in only one species, perhaps studies on other species will allow us to offer a more accurate explanation.

  1. 603-604: Only one study cited, so it must read “This study suggests that  …”

Reply

  1. 603-604: « These studies suggest that » changed to “This study suggests that  …”
  2. 617: Some Cyrillic language is present in this line. Translate or remove!

 Reply

ok

Conclusions:

The first two sentences of the conclusion section emphasize that new insights are provided. Well, this is somewhat relative as descriptions of the serotonergic and dopaminergic neurons in other species exist. That they control motor circuits and behavior is not new.

Reply

deleted«, for the first timе  »

Again, the processing of olfactory as well as mechanosensory and chemosensory input by 5-HT and DA is worthwhile to be outlined, but the optic input is equally important. This, however, is not mentioned. Would the authors take the position in saying the development of the ON indicates that the olfactory input is more important, whilst not having analyzed the optic neuropils?

Reply

Relevant changes have been made

  1. L. 673: Use abbreviations consistently! Here the authors change between 5-HT and DA, the use “serotonin or dopamine” and again “5-HT and DA”. Please take care to check the entire manuscript for such inconsistencies.

Reply

they believed it, deleted it

  1. 647: I think the reference to Barnacles is not really pertinent, as these undergo a more profound metamorphosis and attach, which is not the case with King crabs.

Reply

The link was removed from the text and from the list of references

  1. 678-680: Can the authors become a little more specific on the electrophysical and pharmacological approaches?

Reply

The sentence has been deleted

Reviewer 3 Report

Comments and Suggestions for Authors

Authors have described for the first time the developing serotonin and dopamine neuronal networks in the median brain and VNC of larval and adult P. camtschaticus.

Major

1. Authors should further describe the importance of the King Crab pertaining to society, ex. economy or human health and why exactly this organism was selected for further investigation of its nervous system. 

2. Authors should address any parallels in neuronal network development between the P. camtshaticus and human.

3. While the authors have used CLSM as the method for elucidating various features of serotonin and dopamine localization through development, they have not quantified the measures using statistical analysis.  Example, graphs corresponding to size differences should be plotted where possible and analyzed by means of statistical analysis.  

4. To summarize the findings, at the end of the article in a final Figure, the authors should summarize the findings on serotonergic and dopaminergic neuronal development across various stages from larva to adult as it relates to behavioral patterns/changes in the King crab.  This will make it visually easier for readers to take away the key findings of the study.

Minor

1. Line 35 has a typo in the word "in".

Author Response

Authors have described for the first time the developing serotonin and dopamine neuronal networks in the median brain and VNC of larval and adult P. camtschaticus.

Major

Authors should further describe the importance of the King Crab pertaining to society, ex. economy or human health and why exactly this organism was selected for further investigation of its nervous system. 

Reply

As one of the largest crustaceans in the Far East, the Kamchatka crab is of great commercial importance and is an important object of fishing. The Kamchatka crab is an important part of the marine ecosystem. It has an impact on biological diversity, including the nutrition of other organisms and the maintenance of balance in food chains. The Kamchatka crab is an object of scientific research, which allows us to expand knowledge about its behavior, health and interaction with other organisms.Authors should address any parallels in neuronal network development between the P. camtshaticus and human.

Reply

Serotonin and dopamine is acknowledged as major neuromodulators of nervous systems involved in normal and abnormal behaviors in both invertebrates and vertebrates. Simple model systems, such as crustaceans, are often more amenable than vertebrates for studying mechanisms underlying behaviors. Although various cellular responses of biogenic amines have been characterized in crustaceans, the mechanisms linking these molecules to behavior remain largely unknown. Observed effects of agonists and antagonists DA и 5-НТ in abdomen posture, escape responses, and fighting have led to the suggestion that эти biogenic amine may play a role in modulating interactive behaviors. The influence of serotonin on behavior and physiology has been shown to overlap with that of dopamine. These data indicate that a need to study topographies of specific neuron classes and for more detailed studies of their maturation during the course of larval development. Identified neural circuits are now being used to better understand how the modulatory effects of dopamine and serotonin interact at the single cell level and specific behavior is affected

  1. While the authors have used CLSM as the method for elucidating various features of serotonin and dopamine localization through development, they have not quantified the measures using statistical analysis.  Example, graphs corresponding to size differences should be plotted where possible and analyzed by means of statistical analysis.

Reply

Indeed, we did not keep records of the number of serotonin and dopamine neurons in the departments of the central nervous system. We were not faced with such a task. We focused on the detection of different types of neurons in crab larvae. Nevertheless, the remark is true and we will definitely conduct a quantitative analysis of neurons in brain regions when the general morphology of the crab's central nervous system is understood

  1. To summarize the findings, at the end of the article in a final Figure, the authors should summarize the findings on serotonergic and dopaminergic neuronal development across various stages from larva to adult as it relates to behavioral patterns/changes in the King crab.  This will make it visually easier for readers to take away the key findings of the study.

Reply

The results of the study are summarized in the conclusion.

Minor

  1. Line 35 has a typo in the word "in".

Reply

Done